# Complementary contributions of basolateral amygdala and orbitofrontal cortex to value learning under uncertainty

Alexandra Stolyarova[1]*, Alicia Izquierdo[1,2,3,4]*

[1]Department of Psychology, University of California, Los Angeles, Los Angeles, United States; [2]Integrative Center for Learning and Memory, University of California, Los Angeles, Los Angeles, United States; [3]Integrative Center for Addictions, University of California, Los Angeles, Los Angeles, United States; [4]The Brain Research Institute, University of California, Los Angeles, Los Angeles, United States

**Abstract** We make choices based on the values of expected outcomes, informed by previous experience in similar settings. When the outcomes of our decisions consistently violate expectations, new learning is needed to maximize rewards. Yet not every surprising event indicates a meaningful change in the environment. Even when conditions are stable overall, outcomes of a single experience can still be unpredictable due to small fluctuations (i.e., *expected uncertainty*) in reward or costs. In the present work, we investigate causal contributions of the basolateral amygdala (BLA) and orbitofrontal cortex (OFC) in rats to learning under expected outcome uncertainty in a novel delay-based task that incorporates both predictable fluctuations and directional shifts in outcome values. We demonstrate that OFC is required to accurately represent the distribution of wait times to stabilize choice preferences despite trial-by-trial fluctuations in outcomes, whereas BLA is necessary for the facilitation of learning in response to surprising events.

*For correspondence:
astolyarova@psych.ucla.edu (AS);
aizquie@psych.ucla.edu (AI)

**Competing interests:** The authors declare that no competing interests exist.

## Introduction

Learning to predict rewards is a remarkable evolutionary adaptation that supports flexible behavior in complex and unstable environments. When circumstances change, previously-acquired knowledge may no longer be informative and the behavior needs to be adapted to benefit from novel opportunities. Frequently, alterations in environmental conditions are not signaled by external cues and can only be inferred from deviations from anticipated outcomes, that is, *surprise* signals.

When making decisions, humans typically attempt to maximize benefits (i.e., amount of reward) received per invested resource (i.e., money, time, physical or cognitive effort). We, like many other animals, compute economic value that takes into account rewards and costs associated with available behavioral options and choose the alternative that is expected to result in outcomes of the highest value based on previous experiences under similar conditions (*Padoa-Schioppa and Schoenbaum, 2015*; *Sugrue et al., 2005*). When the outcomes of choices consistently violate expectations, new learning is needed to maximize reward procurement. However, not every unexpected outcome is caused by meaningful changes in the environment. Even when conditions are stable overall, outcomes of a single experience can still be unpredictable due to small fluctuations (i.e., *expected uncertainty*) in reward and costs. Such fluctuations complicate surprise-driven learning since animals need to distinguish between true changes in the environment from stochastic feedback under otherwise stable conditions, known as the problem of change-point detection (*Courville et al., 2006*; *Dayan et al., 2000*; *Gallistel et al., 2001*; *Pearce and Hall, 1980*; *Yu and Dayan, 2005*).

**eLife digest** Nobody likes waiting – we opt for online shopping to avoid standing in lines, grow impatient in traffic, and often prefer restaurants that serve food quickly. When making decisions, humans and other animals try to maximize the benefits by weighing up the costs and rewards associated with a situation. Many regions in the brain help us choose the best options based on quality and size of rewards, and required waiting times. Even before we make decisions, the activity in these brain regions predicts what we will choose.

Sometimes, however, unexpected changes can lead to longer waiting times and our preferences suddenly become less desirable. The brain can detect such changes by comparing the outcomes we anticipate to those we experience. When the outcomes are surprising, specific areas in the brain such as the amygdala and the orbitofrontal cortex help us learn to make better choices. However, as surprising events can occur purely by chance, we need to be able to ignore irrelevant surprises and only learn from meaningful ones. Until now, it was not clear whether the amygdala and orbitofrontal cortex play specific roles in successfully learning under such conditions.

Stolyarova and Izquierdo trained rats to select between two images and rewarded them with sugar pellets after different delays. If rats chose one of these images they received the rewards after a predictable delay that was about 10 seconds, while choosing the other one produced variable delays – sometimes the time intervals were either very short or very long. Then, the waiting times for one of the alternatives changed unexpectedly. Rats with healthy brains quickly learned to choose the option with the shorter waiting time.

Stolyarova and Izquierdo repeated the experiments with rats that had damage in a part of the amygdala. These rats learned more slowly, particularly when the variable option changed for the better. Rats with damage to the orbitofrontal cortex failed to learn at all. Stolyarova and Izquierdo then examined the rats' behavior during delays. Rats with damage to the orbitofrontal cortex could not distinguish between meaningful and irrelevant surprises and always looked for the food pellet (i.e. anticipated a reward) at the average delay interval.

These findings highlight two brain regions that help us distinguish meaningful surprises from irrelevant ones. A next step will be to examine how the amygdala and orbitofrontal cortex interact during learning and see if changes to the activity of these brain regions may affect responses. Advanced methods to non-invasively manipulate brain activity in humans may help people who find it hard to cope with changes; or individuals suffering from substance use disorders, who often struggle to give up drugs that provide them immediate and predictable rewards.

Both the basolateral amygdala (BLA) and orbitofrontal cortex (OFC) participate in flexible reward-directed behavior. Representations of expected outcomes can be decoded from both brain regions during value-based decision making (*Conen and Padoa-Schioppa, 2015*; *Haruno et al., 2014*; *Padoa-Schioppa, 2007*, *2009*; *Salzman et al., 2007*; *van Duuren et al., 2009*). Amygdala lesions render animals unable to adaptively track changes in reward availability or benefit from profitable periods in the environment (*Murray and Izquierdo, 2007*; *Salinas et al., 1996*; *Salzman et al., 2007*). Furthermore, a recent evaluation of the accumulated literature on BLA in appetitive behavior suggests that this region integrates both current reward value and long-term history information (*Wassum and Izquierdo, 2015*), and therefore may be particularly well-suited to guide behavior when conditions change. Importantly, single-unit responses in BLA track surprise signals (*Roesch et al., 2010*) that can drive learning.

Similarly, a functionally-intact OFC is required for adaptive responses to changes in outcome values (*Elliott et al., 2000*; *Izquierdo and Murray, 2010*; *Murray and Izquierdo, 2007*). Impairments produced by OFC lesions have been widely attributed to diminished cognitive flexibility or inhibitory control deficits (*Bari and Robbins, 2013*; *Dalley et al., 2004*; *Elliott and Deakin, 2005*; *Winstanley, 2007*). However, this view has been challenged recently by observations that selective medial OFC lesions cause potentiated switching between different option alternatives, rather than a failure to disengage from previously acquired behavior (*Walton et al., 2010*, *2011*). Indeed, there is increasing evidence that certain sectors of OFC may not exert a canonical inhibitory control over

action, but may instead contribute outcome representations predicted by specific cues in the environment and update expectations in response to surprising feedback (*Izquierdo et al., 2017*; *Marquardt et al., 2017*; *Riceberg and Shapiro, 2012*, *2017*; *Rudebeck and Murray, 2014*; *Stalnaker et al., 2015*).

Despite important contributions of both the BLA and OFC to several forms of adaptive value learning, some learning tasks progress normally without the recruitment of these brain regions. For example, the OFC is not required for acquisition of simple stimulus-outcome associations, both in Pavlovian and instrumental context, or for unblocking driven by differences in value when outcomes are certain and predictable. However, the OFC is needed for adaptive behavior that requires integration of information from different sources, particularly when current outcomes need to be compared with a history in a different context (or state) as in devaluation paradigms (*Izquierdo et al., 2004*; *McDannald et al., 2011*, *2005*; *Stalnaker et al., 2015*). Similarly, as has been shown in rats, BLA has an important role in early learning or decision making under ambiguous outcomes (*Hart and Izquierdo, 2017*; *Ostrander et al., 2011*), and seems to play a limited role in choice behavior when these outcomes are known or reinforced through extended training. These observations hint at important roles for BLA and OFC in learning under conditions of uncertainty. Yet little is known about unique contributions of these brain regions to value learning when outcomes are fluctuating even under stable conditions (i.e., when there is expected uncertainty in outcome values). Furthermore, the functional dissociation between different OFC subregions (e.g. ventromedial vs. lateral) is presently debated (*Dalton et al., 2016*; *Elliott et al., 2000*; *Morris et al., 2016*).

Recently-developed computational models based on reinforcement learning (RL) (*Diederen and Schultz, 2015*; *Khamassi et al., 2011*; *Preuschoff and Bossaerts, 2007*) and Bayesian inference principles (*Behrens et al., 2007*; *Nassar et al., 2010*) are well suited to test for unique contributions of different brain regions to value learning under uncertainty. These models rely on learning in response to surprise, or the deviation between expected and observed outcomes (i.e., *reward prediction errors, RPEs*); the *learning rate*, in turn, determines the degree to which prediction errors affect value estimates. Importantly, the RL principles do not only account for animal behavior, but are also reflected in underlying neuronal activity (*Lee et al., 2012*; *Niv et al., 2015*).

In the present work, we first developed a novel delay-based behavioral paradigm to investigate the effects of expected outcome uncertainty on learning in rats. We demonstrated that rats can detect true changes in outcome values even when they occur against a background of stochastic feedback. Such behavioral complexity in rodents allowed us to assess causal contributions of the BLA and OFC to value learning under expected outcome uncertainty. Specifically, we examined the neuroadaptations that occur in these brain regions in response to experience with different levels of environmental uncertainty and employed fine-grained behavioral analyses partnered with computational modeling of trial-by-trial performance of OFC- and BLA-lesioned animals on our task that incorporates both predictable fluctuations and directional shifts in outcome values.

## Results

### Rats can detect true changes in values despite variability in outcomes

Our delay-based task was designed to assess animals' ability to detect true changes in outcome values (i.e., upshifts and downshifts) even when they occur against the background of stochastic feedback under baseline conditions (expected uncertainty). To probe the effects of expected outcome uncertainty on learning in rodents, we first presented a group of naïve rats (n = 8) with two choice options identical in average wait time but different in the variance of the outcome distribution. Each response option was associated with the delivery of one sugar pellet after a delay interval. The delays were pooled from distributions that were identical in mean, but different in variability (low vs high: LV vs HV; $\sim N(\mu, \sigma)$: $\mu = 10$ s, $\sigma_{HV}$=4s $\sigma_{LV}$=1 s). Following the establishment of stable performance (defined as no statistical difference in any of the behavioral parameters across three consecutive testing sessions, including choice and initiation omissions, average response latencies and option preference), rats experienced value upshifts (delay mean was reduced to 5 s with variance kept constant) and downshifts (delay mean was increased to 20 s) on each option independently, followed by return to baseline conditions (*Figure 1A,B*; *Video 1*, *Video 2*). Each shift and baseline phase lasted five 60-trial testing sessions; therefore, the total duration of the main task was 43

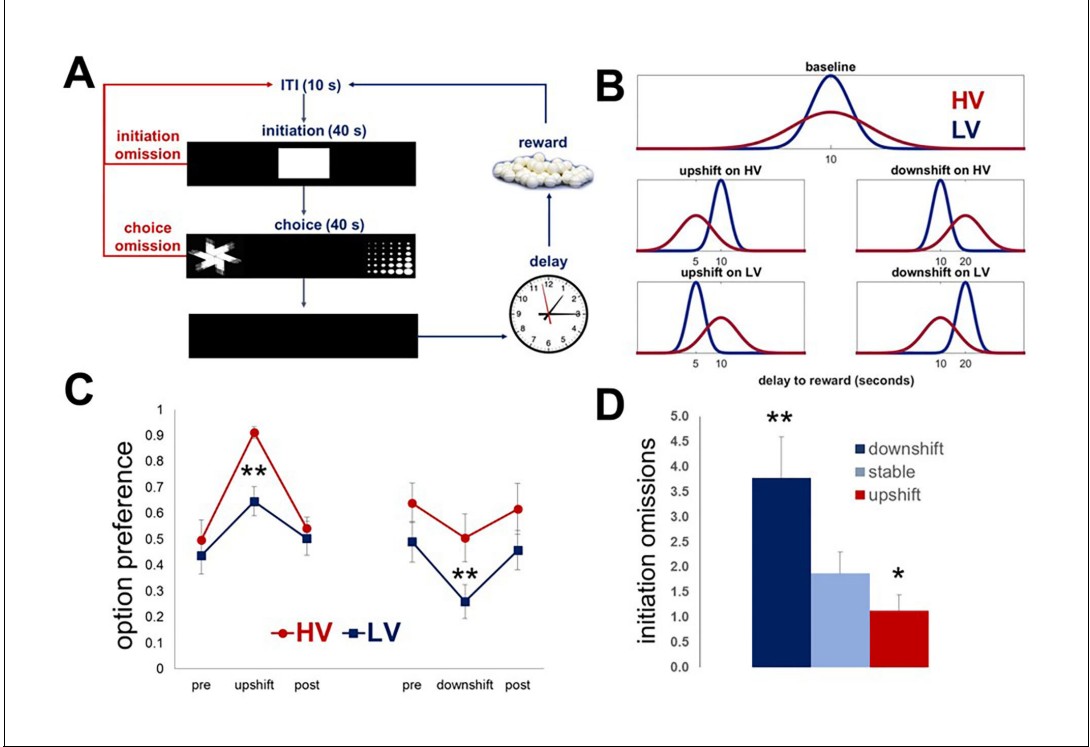

**Figure 1.** Task design and performance of intact animals. Our task is designed to investigate the effects of expected outcome uncertainty on value learning. (A) Each trial began with stimulus presentation in the central compartment of the touchscreen. Rats (n = 8) were given 40 s to initiate a trial. If 40 s passed without a response, the trial was scored as an 'initiation omission.' Following a nosepoke to the central compartment, the central stimulus disappeared and two choice stimuli were presented concurrently in each of the side compartments of the touchscreen allowing an animal a free choice between two reward options. An animal was given 40 s to make a choice; failure to select an option within this time interval resulted in the trial being scored as 'choice omission' and beginning of an ITI. Each response option was associated with the delivery of one sugar pellet after a delay interval. (B) The delays associated with each option were pooled from distributions that are identical in mean value, but different in variability: LV (low variability, shown in blue) vs. HV (high variability, shown in red); ~N($\mu$, $\sigma$): $\mu$ = 10 s, $\sigma_{HV}$=4s, $\sigma_{LV}$=1s. Following the establishment of stable performance, rats experienced value upshifts ($\mu$ = 5 s; $\sigma$ kept constant) and downshifts ($\mu$ = 20 s) on each option independently, followed by return to baseline conditions. Each shift and return to baseline phase lasted for five 60-trial sessions. (C) Regardless of the shift type, animals significantly changed their preference in response to all shifts (all p values<0.05). However, significant differences between HV and LV in choice adaptations were observed for both upshifts and downshifts: greater variance of outcome distribution at baseline facilitated behavioral adaptation in response to value upshifts (HV vs LV difference, p=0.004), but rendered animals suboptimal during downshifts (p=0.027); conversely, low expected uncertainty at baseline led to decreased reward procurement during upshifts in reward. The data are shown as group means for option preference during pre-baseline, shift and post-baseline conditions, ± SEM. The asterisks signify statistical differences between HV and LV conditions. (D) The number of initiation omissions was significantly increased during downshift (p=0.004) and decreased during upshifts (p=0.017) in value, regardless of the levels of expected uncertainty, demonstrating effects of overall environmental reward conditions on motivation to engage in the task. The data are shown as group means by condition +SEM. *p<0.05, **p<0.01. Summary statistics and individual animal data are provided in *Figure 1—source data 1*.

The following source data is available for figure 1:

**Source data 1.** Summary statistics and individual data for naïve animals performing the task.

testing days for each animal. Maximal changes in the choice of each option in response to shifts were analyzed with omnibus within-subject ANOVA with shift type (HV, LV; upshift, downshift) and shift phase (pre-shift baseline, shift, post-shift baseline) as within-subject factors. These analyses identified a significant shift type x phase interaction [F(6, 42)=16.412, p<0.0001]. Post-hoc analyses revealed no differences in preference at baseline conditions across assessments [F(3.08, 21.57)=0.98, p=0.422; Greenhouse-Geisser corrected], suggesting that rats were able to infer mean option values (wait times) and maintain stable choice preferences despite variability in outcomes.

All animals significantly changed their preference in response to all shifts (*Figure 1*, all p values<0.05). We then assessed the effects of the overall environmental reward conditions on rats'

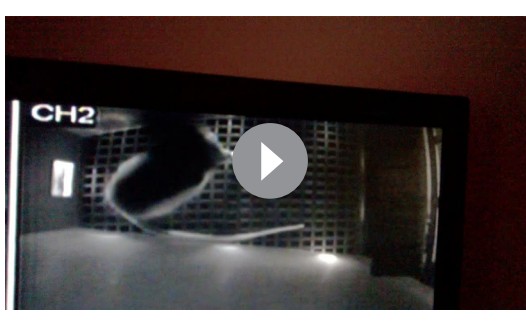

**Video 1.** An animal performing the task during upshift on HV option. During an upshift in value on each option, the mean of the delays to reward was reduced to 5 s with variance kept the same as during baseline conditions.

motivation to engage in the task. The number of initiation omissions (i.e., failure to respond to the central cue presented at the beginning of each trial within 40 s) was analyzed with omnibus ANOVA with reward conditions (stable, upshift, and downshift collapsed across HV and LV options) as within-subject factor. The main effect of condition was significant [$F_{(1.09, 7.61)}=16.772$, $p=0.03$; Greenhouse-Geisser corrected]: the number of omissions was significantly increased during downshifts ($p=0.004$) and decreased during upshifts ($p=0.017$) in value, revealing that task engagement was sensitive to overall environmental reward rate.

Therefore, rodents are able to learn about fundamental directional changes in value means despite stochastic fluctuations in outcome values under baseline conditions (i.e., expected uncertainty). However, significant differences between HV and LV in choice adaptations were observed for both upshifts and downshifts: greater variance of outcome distribution at baseline facilitated behavioral adaptation in response to value upshifts (HV vs LV difference, $p=0.004$), but rendered animals suboptimal during downshifts ($p=0.027$); conversely, low expected uncertainty at baseline led to decreased reward procurement during upshifts in reward. These effects may be explained by a hyperbolic nature of delay-discounting across species (*Freeman et al., 2009*; *Green et al., 2013*; *Hwang et al., 2009*; *Mazur and Biondi, 2009*; *Mitchell et al., 2015*; *Rachlin et al., 1991*).

## Experience with uncertainty induces distinct patterns of neuroadaptations in the BLA and OFC

We hypothesized that experience with different levels of outcome uncertainty would induce long-term neuroadaptations, affecting the response to the same magnitude of surprise signals. Specifically, we assessed expression of gephyrin (a reliable proxy for membrane-inserted GABA$_A$ receptors mediating fast inhibitory transmission; [*Chhatwal et al., 2005*; *Tyagarajan et al., 2011*]) and GluN1 (an obligatory subunit of glutamate NMDA receptors; [*Soares et al., 2013*]) in BLA and OFC. Three separate groups of animals were trained to respond to visual stimuli on a touchscreen to procure a reward after variable delays. The values of outcomes were identical to our task described above but no choice was given. One group was trained under LV conditions, the second under HV (matched in total number of rewards received), and the third control group received no rewards (n = 8 in each group, total n = 24). Given the limited amount of tissue, we focused on NMDA instead of AMPA receptors based on previous evidence demonstrating dissociable effects of ionotropic glutamate receptors in delay-based decision making (*Yates et al., 2015*).

Protein expression analyses revealed unique adaptations to outcome variability in BLA, specifically in GABA-ergic sensitivity. Biochemical measures were analyzed with mixed ANOVA with brain region as a within-subject factor and reward experience (HV, LV or no reward) as a between-subject factor. There was a significant main effect of group [$F_{(2,12)}=6.002$, $p=0.016$] and brain region x group interaction [*Figure 2A*; $F_{(2,12)}=41.863$, $p<0.0001$] for gephyrin. A significant main effect of group [$F_{(2,21)}=4.084$, $p=0.032$] and group x brain region [$F_{(2,21)}=5.291$, $p=0.014$] interaction were also found for

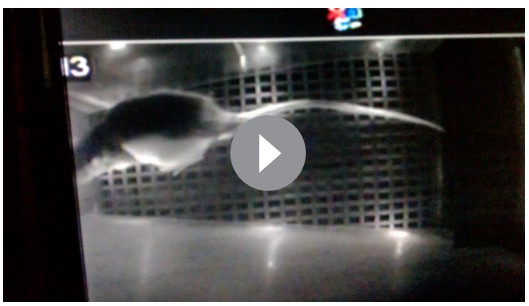

**Video 2.** An animal performing the task during downshift on HV option. During a downshift in value on each option, the mean of the delays to reward was increased to 20 s with variance kept constant.

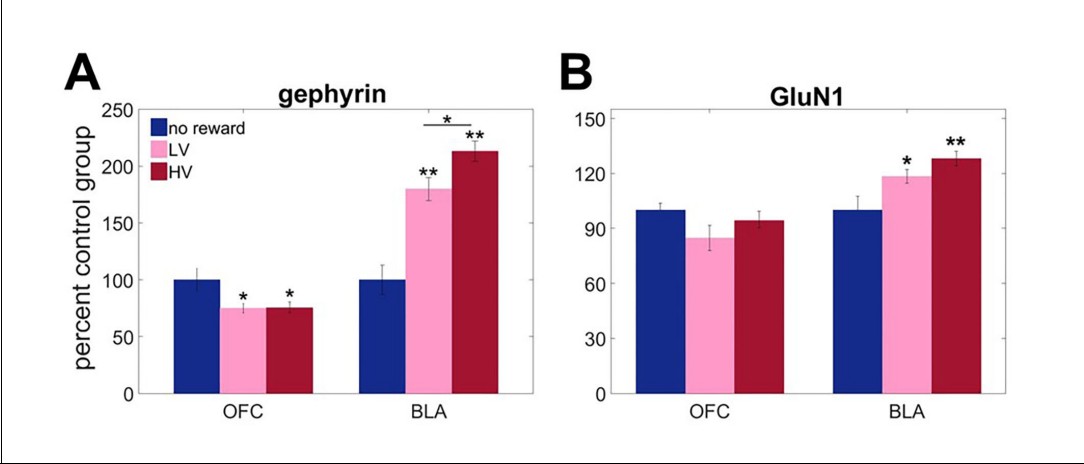

**Figure 2.** Region-specific alterations in gephyrin and GluN1 expression induced by experience with outcome uncertainty. Three separate groups of animals were trained to respond to visual stimuli on a touchscreen to get a reward after variable delays. The values of outcomes were identical to the main task but no choice was given. One group was trained under LV conditions, the second under HV (matched in total number of rewards received), and the third control group received no rewards (n = 8 per group). We assessed expression of **A** gephyrin (a reliable proxy for membrane-inserted GABA$_A$ receptors mediating fast inhibitory transmission) and **B** GluN1 (an obligatory subunit of glutamate NMDA receptors) in BLA and ventral OFC. Biochemical analyses revealed uncertainty-dependent upregulation in gephyrin in BLA, that was maximal following HV training (p<0.0001). Similarly, GluN1 showed robust upregulation in response to experienced reward in BLA (no reward vs LV p=0.045; no reward vs HV p=0.002), however post hoc analyses failed to detect a significant difference between HV and LV training (p=0.637). In ventral OFC, gephyrin was downregulated in response to experiences with reward in general (no reward vs LV p=0.045; no reward vs HV p=0.042) and did not depend on variability in outcome distribution; no changes were observed in GluN1. The data are shown as group means by condition +SEM. *p<0.05, **p<0.01 Summary statistics and individual animal data are provided in *Figure 2—source data 1*.

The following source data is available for figure 2:

**Source data 1.** Summary statistics and individual data for GluN1 and gephyrin expression in BLA and OFC.

GluN1 expression. Subsequent analyses identified uncertainty-dependent upregulation of gephyrin in BLA [between-subject ANOVA: F(2,21)=45.448, p<0.0001), that was maximal following HV training (all post hoc comparison p values<0.05). Similarly, GluN1 showed robust upregulation in response to experienced reward in BLA [*Figure 2B*; F(2,21)=7.092, p=0.004; no reward vs LV p=0.045; no reward vs HV p=0.002], however post hoc analyses failed to detect a significant difference between HV and LV training (p=0.637). In OFC, gephyrin was instead downregulated in response to experiences with reward in general [F(2,12)=4.445, p=0.036; no reward vs LV p=0.045; no reward vs HV p=0.042] and did not depend on variability in outcome distribution (post hoc comparison: HV vs LV, p=1); no changes were observed in GluN1 [F(2,21)=2.359, p=0.119].

Therefore, both the BLA and OFC undergo unique patterns of neuroadaptations in response to experience with variability, suggesting that these brain regions may play complementary, yet dissociable, roles in value learning under outcome uncertainty. Given the behavioral complexity that rodents exhibit on our task, we were able to directly test the causal contributions of the BLA and ventromedial OFC to value learning under conditions of expected uncertainty in outcome distribution.

## Causal contributions of the BLA and OFC to value learning under uncertainty

The results of lesion studies (lesion sites are shown in *Figure 3*) were in line with predictions suggested by protein data. Because we were primarily interested in the contributions of the BLA and OFC to surprise-driven learning, we first analyzed the maximal changes in option preference in response to up- and downshifts. This analysis allowed us to control for potential effects of brain lesions on choice behavior under baseline conditions in our task. An omnibus ANOVA with shift type as within- and experimental group (sham, BLA vs OFC lesion; n = 8 per group; total n = 24) as

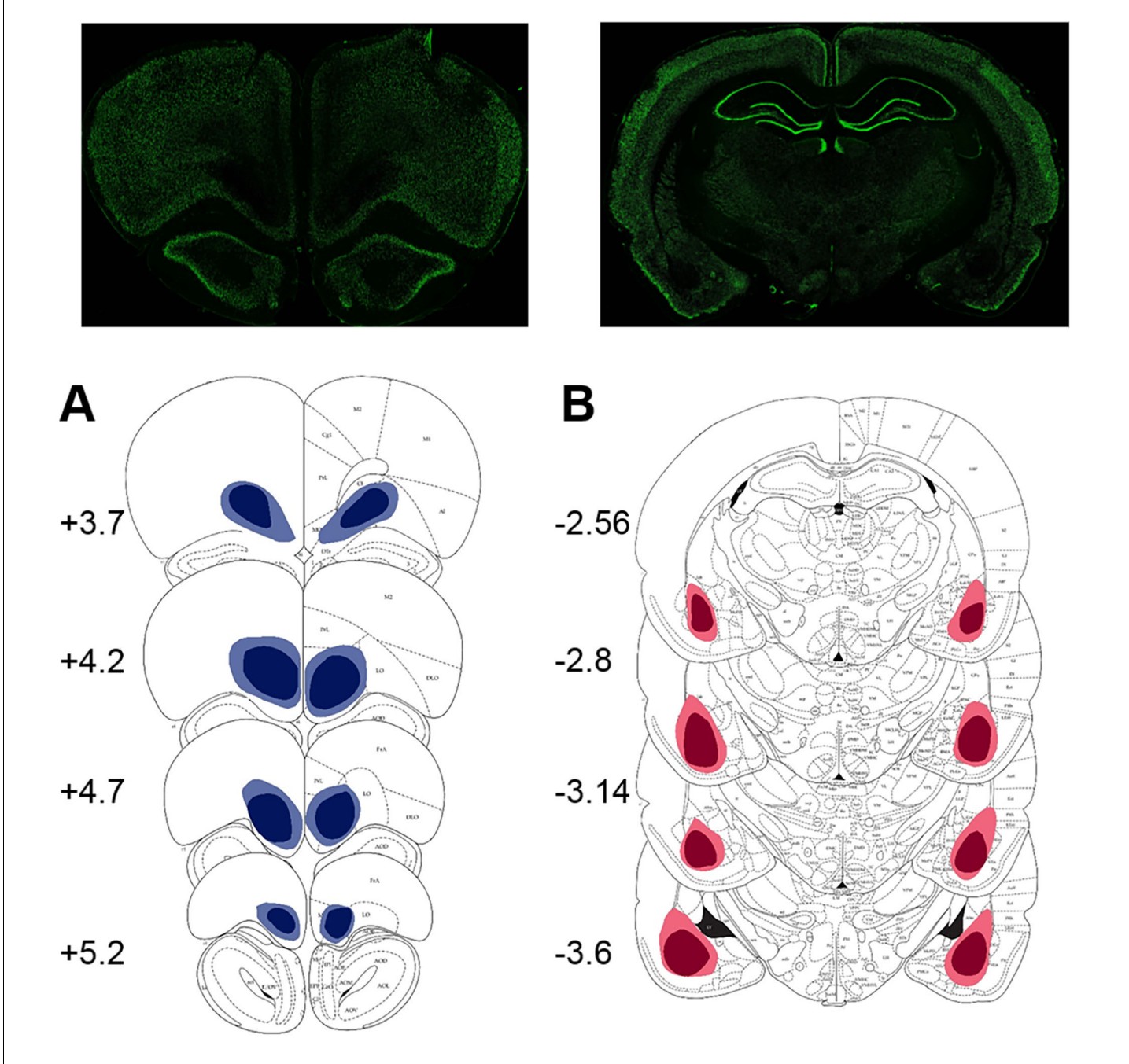

**Figure 3.** Location and extent of intended lesion (colored regions) on standard coronal sections through ventral OFC and BLA. The extent of the lesions was assessed after the completion of behavioral testing by staining for a marker of neuronal nuclei, NeuN. (**A**) Top: representative photomicrograph of a NeuN stained coronal section showing ventral OFC lesion. Bottom: depictions of coronal sections adapted from (*Paxinos and Watson, 1997*). The numerals on the lower left of each matched section represent the anterior-posterior distance (mm) from Bregma. Light and dark blue represent maximum and minimum lesion area across animals, respectively. Though coordinates were aimed at the ventral orbital region, lesion extent includes anterior medial orbital cortex as well. (**B**) Top: representative photomicrograph of a NeuN stained coronal section showing BLA lesion. Bottom: depictions of coronal sections with numerals on the lower left of each matched section representing the anterior-posterior distance (mm) from Bregma. Light and dark red represent maximum and minimum lesion area across animals, respectively.

between-subject factors detected a significant main effect of group [F(2,21)=11.193, p<0.0001] and group x shift type interaction [F(6,63)=9.472, p<0.0001]. Subsequent analyses showed significant simple main effects of experimental group on all shift types: upshift on HV [F(2,21)=14.723, p<0.0001], upshift on LV [F(2,21)=5.663, p=0.011], downshift on HV [F(2,21)=19.081, p<0.0001], and downshift on LV [F(2,21)=7.189, p=0.004]. The OFC-lesioned rats were less optimal on our task: they changed their option preference to a significantly lesser degree compared to control animals during upshifts on HV (p=0.005) and LV (p=0.039), as well as the downshift on LV option (p=0.015; *Figure 4A*). Whereas OFC lesions produced a pronounced impairment in performance, it was less clear if alterations produced by BLA lesions lead to suboptimal behavior. BLA-lesioned animals changed their option preference to a lesser degree on HV upshifts (p<0.0001), but compensated by exaggerated adaptations to HV downshifts (p<0.0001; *Figure 4A*).

In addition to examining the maximal changes in option preferences, we analyzed the behavioral data with an omnibus ANOVA with shift type and shift phase (pre-shift baseline, shift performance, and post-shift baseline) as within-subject and experimental group as between-subject factors. This test similarly detected a significant shift type x phase x group interaction [F(6.9,72.5)=7.41, p<0.0001; Greenhouse-Geisser corrected, *Figure 4—figure supplement 1*). Consistent with the preceding analyses, post hoc tests revealed reduced adaptations to value upifts on the HV option in both lesion groups (p<0.01). However, we also observed more frequent choices of the LV option when its value was increased in BLA-lesioned animals (p<0.01) as well as reduced HV option preference (p<0.01) and increased LV option preference (p<0.05) during downshifts in both lesion groups compared to control animals. This pattern of results may be explained by changes in choice behavior even under baseline conditions in BLA- and OFC-lesioned animals that interacted with rats' ability to learn about shifts in value.

Successful performance on our task required animals to distinguish between the variance of outcome distributions under stable conditions from surprising shifts in value, despite the fact that delay distributions at baseline and distributions during the shift partially overlapped. To evaluate whether the animals in lesioned groups adopted a different strategy and demonstrated altered sensitivity to surprising outcomes, we examined the win-stay/lose-shift responses. Win-stay and lose-shift scores were computed based on trial-by-trial data similar to previous reports (*Faraut et al., 2016*; *Imhof et al., 2007*; *Worthy et al., 2013*): a score of 1 was assigned when animals repeated the choice following better than average outcomes (*win-stay*) or switched to the other alternative following worse than average outcomes (*lose-shift*). Win-shift and lose-stay trials were counted as 0 s. To specifically address whether rats distinguished expected fluctuations from surprising changes, we divided the trials into two types: when the delays fell within distributions experienced for each option at baseline (expected outcomes) and those in which the degree of surprise exceeded that expected by chance. The algorithm used for this analysis kept track of all delays experienced under baseline conditions before the current trial for each animal individually. On each trial, we found the value of the minimal and maximal delay. If the current delay value fell within this interval, the outcome was classified as *expected*. If the current delay fell outside of this distribution, the outcome on this trial was classified as *unexpected* (surprising).

Win-stay and lose-shift scores were calculated for each trial type separately and their probabilities (summary score divided by the number of trials) for both trial types were subjected to ANOVA with strategy as within-subject and experimental group as between-subject factors. Our analyses indicated significant strategy x experimental group interaction [F(6,63)=9.912, p<0.0001]. Critically, sham-lesioned animals demonstrated increased sensitivity to unexpected outcomes compared to predictable fluctuations for both wins and losses (*Figure 4B*, p values <0.0001). Similarly, the ability to distinguish between expected and unexpected outcomes was intact in BLA-lesioned animals (p values < 0.001), although their sensitivity to feedback decreased overall. In contrast, OFC-lesioned animals failed to distinguish predictable from surprising fluctuations. Interestingly, sham and BLA-lesioned animals demonstrated low win-stay and lose-shift scores when trial outcomes were expected; these animals were more likely to shift after better than average outcomes and persist with their choices after worse outcomes. In addition to feedback insensitivity, such behavior may result from increases in exploratory behavior in response to wins and behavioral inflexibility after losses. Additionally, when outcomes are relative stable and predictable, rats may be more sensitive to long-term reward history and rely less on the outcome of any one given trial.

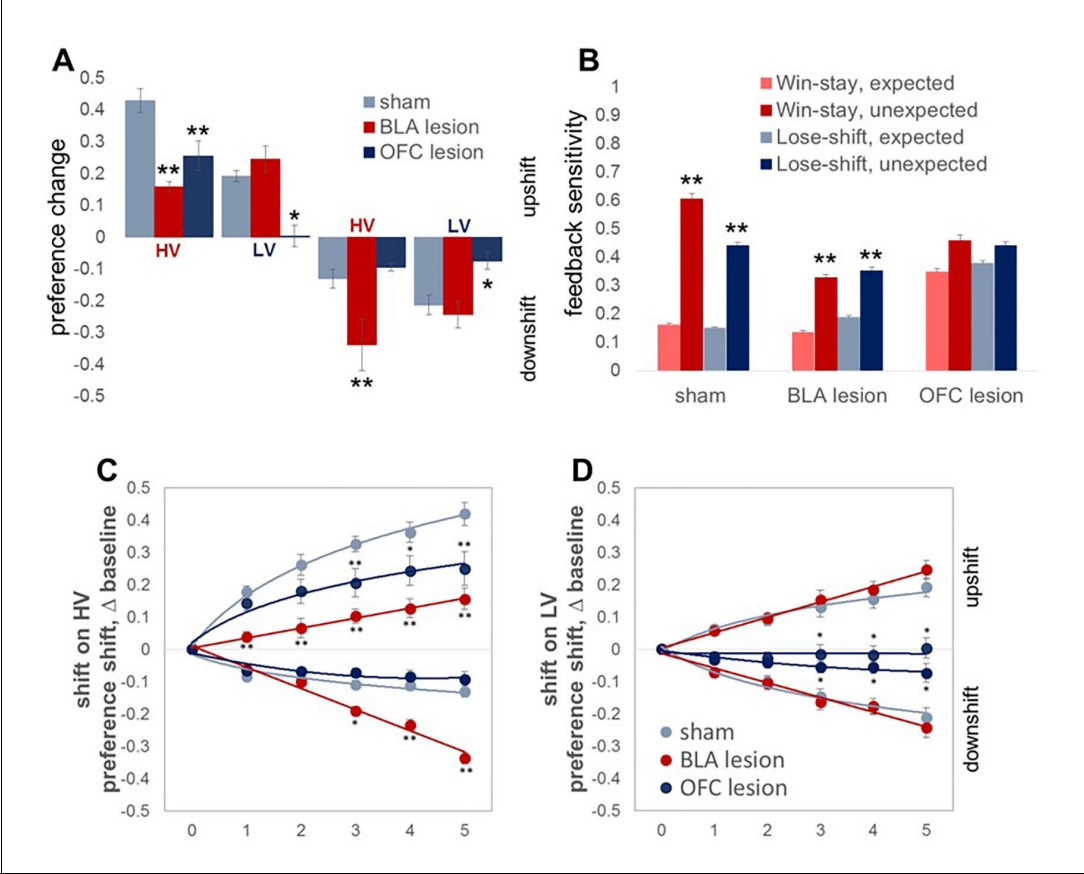

**Figure 4.** Changes in choice preference in response to value shifts and learning strategies in experimental groups. (A) The OFC-lesioned rats (n = 8) were less optimal on our task: they changed their option preference to a significantly lesser degree compared to control animals (n = 8) during upshifts on HV (p=0.005) and LV (p=0.039), as well as the downshift on LV option (p=0.015). Conversely, animals with BLA lesions (n = 8) changed their option preference to a lesser degree on HV upshifts (p<0.0001), but compensated by exaggerated adaptations to HV downshifts (p<0.0001). Group means for option preference during pre-baseline, shift and post-baseline conditions are shown in *Figure 4—figure supplement 1*. We broke the trials into two types: when the delays fell within distributions experienced for each option at baseline (*expected* outcomes) and those in which the degree of surprise exceeded that expected by chance (*unexpected* outcomes). Win-stay/lose-shift scores were computed based on trial-by-trial data: a score of 1 was assigned when animals repeated the choice following better than average outcomes (*win-stay*) or switched to the other alternative following worse than average outcomes (*lose-shift*). Sham-lesioned animals demonstrated increased sensitivity to unexpected feedback (p values < 0.001). Similarly, the ability to distinguish between expected and unexpected outcomes was intact in BLA-lesioned animals (p values < 0.001), although their sensitivity to feedback decreased overall. In contrast, OFC-lesioned animals failed to distinguish expected from unexpected fluctuations. (C,D) To examine the learning trajectory we analyzed the evolution of option preference. BLA-lesioned animals were indistinguishable from controls during the shifts on LV option. Whereas, this experimental group demonstrated significantly attenuated learning during the upshift on HV (p values < 0.0001 for all sessions) and potentiated performance during sessions 3 through 5 on HV downshift (p values < 0.05) compared to sham group. Conversely, learning in OFC-lesioned animals was affected on the majority of the shift types: these animals demonstrated significantly slower learning during sessions 3 through 5 during upshift on HV (p values < 0.05), all sessions during upshift on LV (p values < 0.05) and sessions 3 through 5 during downshift on LV (p values < 0.05). Session 0 refers to baseline/pre-shift option preference. Despite these differences in responses to shifts in value under conditions of uncertainty, we did not observe any deficits in basic reward learning in either the BLA- or OFC-lesioned animals, shown in *Figure 4—figure supplement 2*. The data are shown as group means by condition +SEM. *p<0.05, **p<0.01. Summary statistics and individual animal data are provided in *Figure 4—source data 1* and *Figure 4—source data 2*.

The following source data and figure supplements are available for figure 4:

**Source data 1.** Summary statistics and individual data for changes in choice preference and learning strategies.

**Source data 2.** Summary statistics and individual data demonstrating experimental group differences in response to shifts.

**Figure supplement 1.** Changes in choice behavior in response to value shifts.

**Figure supplement 2.** The lack of group differences in basic reward learning.

## Lesions to the BLA and OFC alter learning trajectory

To examine the learning trajectory we analyzed the evolution of option preference during shift conditions. Specifically, we subjected the session-by-session data during each swift to an omnibus ANOVA with testing session (1 through 5; session 0 in *Figure 4C,D* corresponds to pre-shift option preference) and shift type as within- and experimental group as between subject factors. This analysis revealed a three-way session x shift type x group interaction [$F_{(8.73, 91.71)}=8.418$, $p=0.002$; Greenhouse-Geisser corrected, *Figure 4C,D*]. Subsequent analyses identified significant two-way session x group interactions for each shift type [upshift on HV: $F_{(5.24, 55.04)}=3.585$, $p=0.006$; downshift on HV: $F_{(4.14, 43.452)}=25.646$, $p<0.0001$; upshift on LV: $F_{(2.59, 27.14)} = 4.378$, $p=0.016$; downshift on LV: $F_{(3.69, 38.767)}=6.768$, $p<0.0001$; all Greenhouse-Geisser corrected]. BLA-lesioned animals were indistinguishable from controls during the shifts on LV option. However, this experimental group demonstrated significantly attenuated learning during the upshift on HV (p values < 0.0001 for all sessions) and potentiated performance during sessions 3 through five during the downshifts on HV (p values < 0.05) compared to the sham group. Conversely, learning in OFC-lesioned animals was affected on the majority of the shift types: these animals demonstrated significantly slower learning during sessions 3 through five during upshift on HV (p values < 0.05), all sessions during upshift on LV (p values < 0.05) and sessions 3 through five during downshift on LV (p values < 0.05).

Despite these differences in responses to shifts in value, we did not observe any deficits in basic reward learning in either the BLA- or OFC-lesioned animals. Our surgeries took place prior to any exposure to the testing apparatus or behavioral training, yet both lesioned groups were indistinguishable from controls at early stages of the task. All animals took a similar number of days to learn to nosepoke visual stimuli on the touchscreen to receive sugar rewards [$F_{(2,21)}=0.231$, $p=0.796$] and to initiate a trial [$F_{(2,21)}=0.199$, $p=0.821$]. Similarly, there were no group differences in their responses to the introduction of a 5 s delay interval during pre-training [$F_{(2,21)}=0.679$, $p=0.518$] or the number of sessions to reach stable performance during the initial baseline phase of our uncertainty task [$F_{(92,21)}=0.262$, $p=0.772$; *Figure 4—figure supplement 2*].

## Complementary contributions of the BLA and OFC to value learning under uncertainty revealed by computational modeling

We fit different versions of RL models to trial-by-trial choices for each animal separately. Specifically, we considered the standard Rescorla-Wagner model (RW) and a dynamic learning rate model (Pearce-Hall, PH). The RW model updates option values in response to RPEs (i.e., the degree of surprise) with a constant learning rate, conversely the PH model allows for learning facilitation with surprising feedback (i.e., the learning rate is scaled according to absolute prediction errors). We also compared models in which expected outcome uncertainty is learned simultaneously with value and scales the impact of prediction errors on value (RW+expected uncertainty) and learning rate (Full model) updating. The total number of free parameters, BIC and parameter values for each model and experimental group are provided in *Table 1*. The behavior of the control group was best captured by the dynamic learning rate model with RPE scaling proportional to expected outcome uncertainty and facilitation of learning in response to surprising feedback (Full model; *Table 1*, lower BIC values indicate better fit). Therefore, rats in our experiment increased learning rates in response to surprise to maximize reward acquisition rate, but only if unexpected outcomes were not likely to result from value fluctuations under otherwise stable conditions. Consistent with attenuated learning observed in animals with BLA lesions, trial-by-trial performance in these animals was best fit by RW +expected uncertainty model, demonstrating selective loss of learning potentiation in response to surprise and preserved RPE scaling with expected uncertainty in these animals, leading to slower learning compared to intact animals during the shifts on HV option. Conversely, performance of OFC-lesioned animals was best accounted for by PH model, suggesting that while these animals still increased learning rates in response to surprise, they were insensitive to expected outcome uncertainty. Furthermore, the overall learning rates were reduced in OFC-lesioned animals (p=0.01 compared to the sham group). Finally, we observed significantly lower values of $\beta$ (inverse temperature parameter in softmax choice rule) in both BLA- and OFC-lesioned animals [$F_{(2,21)}=4.88$, $p=0.018$; sham vs BLA: $p<0.0001$; sham vs OFC: $p<0.0001$], suggesting that their behavior is less stable, more

**Table 1.** Model comparison. Lower BIC values indicate better model fit (in bold); number of free parameters and parameter values ± SEM of the best fitting model are provided for each group. Trial-by-trial choices of the intact animals were best captured by the dynamic learning rate model incorporating RPE scaling proportional to expected uncertainty and facilitation of learning in response to surprising outcomes (Full model). BLA lesions selectively eliminated learning rate scaling in response to surprise (RW+expected uncertainty model provided the best fit). Whereas OFC lesioned animals still increased learning rates in response to surprising events (PH model), RPE scaling proportional to expected outcome uncertainty was lost in this group. Furthermore, the overall learning rates were reduced in OFC-lesioned animals (p=0.01). Finally, we observed significantly lower values of $\beta$ (inverse temperature parameter in soft-max choice rule) in both BLA- and OFC-lesioned animals (p<0.0001), suggesting that their behavior is less stable, more exploratory and less dependent on the difference in learned outcome values. Asterisks indicate parameter values that were significantly different from the control group (in bold).

| Model | RW | PH | RW+expected uncertainty | Full | | | | | |
|---|---|---|---|---|---|---|---|---|---|
| # parameters | 3 | 4 | 5 | 6 | | | | | |
| | BIC | | | parameter value ± SEM | | | | | |
| | | | | k | $\alpha$, value | $\beta$ | $\eta$ | $\alpha$, risk | $\omega$ |
| sham | 26519.39 | 26900.66 | 26384.18 | *25681.7* | 0.29 ± 0.03 | 0.09 ± 0.01 | 14.1 ± 0.99 | 0.33 ± 0.04 | 0.56 ± 0.08 | 3.04 ± 0.11 |
| BLA lesion | 26201.89 | 26864.74 | *25153.82* | 27162.82 | 0.32 ± 0.02 | 0.07 ± 0.01 | *7.4 ± 0.6** | n/a | 0.58 ± 0.06 | 3.40 ± 0.4 |
| OFC lesion | 24292.54 | *23171.46* | 24630.92 | 23994.5 | 0.3 ± 0.05 | *0.05 ± 0.01** | *5.5 ± 0.68** | 0/32 ± 0.05 | n/a | n/a |

exploratory and less dependent on the difference in learned outcome values compared to control group.

## Animals with ventral OFC lesions fail to represent expected uncertainty in wait time distributions

To gain further insights into outcome representations in our experimental groups, we analyzed the microstructure of rats' choice behavior. Specifically, we addressed whether BLA and ventral OFC lesions altered animals' ability to form expectations about the timing of reward delivery. On each trial during all baseline conditions, where the overall values of LV and HV options were equivalent, reward port entries were recorded in 1 s bins during the waiting period (after a rat had indicated its choice and until reward delivery; histograms of true distributions of the delays and animals' reward-seeking actions normalized to the total number of reward port entries are shown in *Figure 5*). These data were analyzed with an ANOVA with time bin as within- and lesion group as between-subject factors. There were no significant differences in the mean of expected reward delivery times across groups [$F_{(5,42)}$=1.064, p=0.394]. Similarly, all groups were matched in the total number of reward port entries [$F_{(2,21)}$=0.462, p=0.636; *Figure 5—figure supplement 1*]. However, a significant difference in variances of reward port entry distributions was detected [$\chi^2_{(209)}$=4004.054, p<0.0001]. Whereas the distributions of reward-seeking times in BLA-lesioned rats were indistinguishable from control animals' and the true delays, OFC-lesioned animals concentrated their reward port entries in the time interval corresponding to mean delays, suggesting that while these animals can infer the average outcomes, they fail to represent the variance (i.e., expected uncertainty).

We also considered the changes in waiting times across our task. We calculated the variance of reward port entry times during each baseline (initial phase of the task and four baseline separating the shifts) for each animal. We then subjected the estimated variances to ANOVAs with baseline order (1st to 5th) as within- and lesion group as between-subject factors. Similar to our preceding analysis of combined baselines, we did not detect any group differences in waiting times for the LV option (all p values>0.2). However, there was a significant main effect of lesion group on waiting time variances for the HV option [$F_{(2,21)}$=117.074, p<0.0001; *Figure 5—figure supplement 1*] with OFC-lesioned animals demonstrating consistently lower variability in their waiting behavior despite the experience with shifts. Importantly, since our analyses only included the waiting time prior to reward delivery, these results suggest that OFC-lesioned animals retain the ability to form simple outcome expectations based on long-term experience, yet their ability to represent the more complex outcome distributions is compromised.

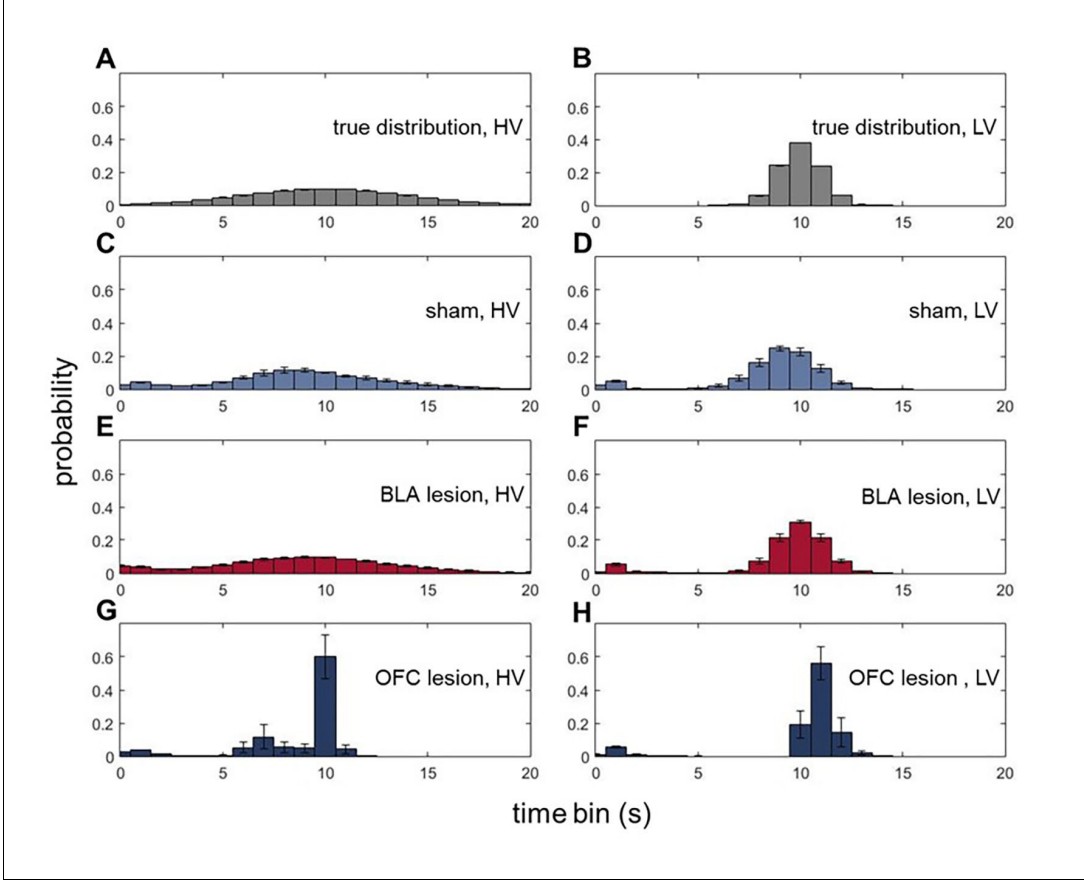

**Figure 5.** Animals with ventral OFC lesions fail to represent expected uncertainty in reward delays. We assessed whether BLA and ventral OFC lesions alter animals' ability to form expectations about the timing of reward delivery. On each trial during all baseline conditions where the overall value of LV and HV options were equivalent, reward port entries were recorded in 1 s bins during the waiting period. There were no significant differences in the means of expected reward delivery times across groups (p=0.394). Similarly, the groups were matched in the total number of reward port entries (p=0.636) as shown in *Figure 5—figure supplement 1*. Whereas the distributions of reward-seeking times in BLA-lesioned animals were indistinguishable from control animals' and the true delays (A–F), OFC-lesioned animals concentrated their reward port entries in the time interval corresponding to mean delays (G,H), suggesting that while these animals can infer the average outcome, they fail to represent the variance (i.e., expected uncertainty). We also considered the changes in waiting times across our task; these data are shown in *Figure 5—figure supplement 1*. Each bar in histogram plots represents mean frequency normalized to total number of reward port entries ±SEM.

The following figure supplement is available for figure 5:

**Figure supplement 1.** Total number of reward port entries and changes in waiting time variances across task phases.

## Lesions to the BLA and ventral OFC induce an uncertainty-avoidant phenotype under baseline conditions

To assess group differences in uncertainty-seeking or avoidance, we subjected HV option preference data under baseline conditions to an ANOVA with time (five repeated baseline tests separating the value shifts) as within- and lesion group as between-subject factors. In addition to their effects on value learning, lesions to both the BLA and ventral OFC induced an uncertainty-avoidant phenotype with animals in both experimental groups demonstrating reduced preference for the HV option under baseline conditions compared to the control group at the beginning of testing [time x group interaction: $F_{(4.37,45.87)} = 8.484$, p<0.0001; post hoc sham vs BLA: p=0.002; sham vs OFC:

p=0.002, *Figure 6*]. BLA-lesioned animals continued to avoid the uncertain option for the entire duration of our experiment (all p values < 0.05, except for baseline three assessment when this group was not different from control animals). However, OFC-lesioned animals increased their choices of the HV option during baseline conditions with repeated testing: they were indistinguishable from controls during baselines 3 and 4 and even demonstrated a trend for higher preference than the control group during the last baseline [post hoc test, OFC vs sham: p=0.059].

## Discussion

Volatile reward statistics were one of the central characteristics of ancestral habitats, favoring the selection of behavioral phenotypes that are able to cope with uncertainty (*Emery, 2006*; *Potts, 2004*; *Steppan et al., 2004*). Most mammals are able to learn higher-order statistics of the environment (*Cikara and Gershman, 2016*; *Gershman and Niv, 2010*; *Niv et al., 2015*) and optimize learning rates based on the degree of uncertainty (*Behrens et al., 2007*; *Nassar et al., 2010*; *Payzan-LeNestour and Bossaerts, 2011*). Until recently, most of the studies have been carried out in the context of probabilistic feedback, where stochasticity in outcomes is driven by reward

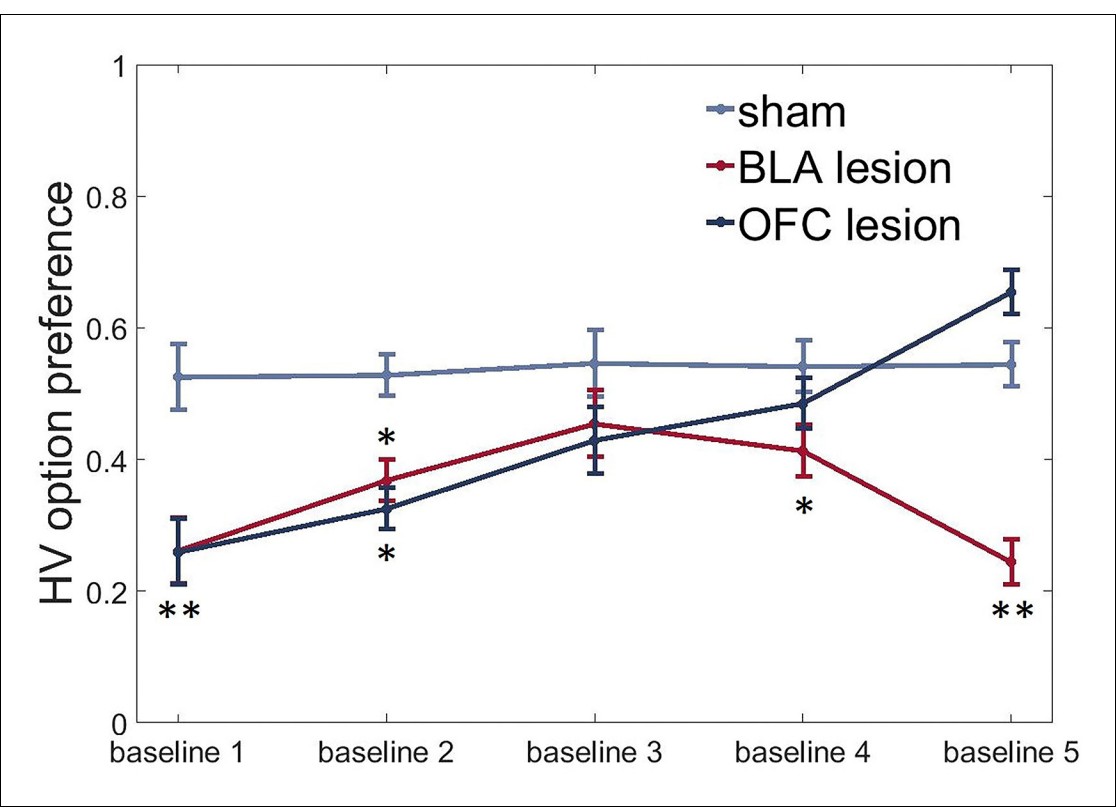

**Figure 6.** BLA and ventral OFC lesions induce uncertainty-avoidance. We observed significantly reduced preference for the HV option under baseline conditions in both experimental groups compared to control animals at the beginning of testing (sham vs BLA: p=0.002; sham vs OFC: p=0.002). BLA-lesioned animals continued to avoid the risky option for most of the experiment (all p values < 0.05, except for baseline three assessment when this group was not different from controls). OFC-lesioned animals progressively increased their choices of HV option during baseline conditions with repeated testing: they were indistinguishable from controls during baselines 3 and 4 and even demonstrated a trend for higher preference than control group during the last baseline [post hoc test, OFC vs sham: p=0.059]. The data are shown as group means by condition ±SEM, *p<0.05, **p<0.01. Summary statistics and individual animal data are provided in *Figure 6—source data 1*.

The following source data is available for figure 6:

**Source data 1.** Summary statistics and individual data for HV option preference following lesions.

omission in a subset of trials. Unlike in laboratory tasks, uncertainty in naturalistic settings is not limited to probabilistic binary outcomes, but also includes variability in delays and effortful costs required to obtain the desired rewards. In the present work, we developed a delay-based task for rats to investigate the effects of expected outcome uncertainty on value learning. Our results provide the first evidence that rats can detect and learn about the true changes in outcome values even when they occur against a background of stochastic delay costs. In our task, animals successfully changed their choice behavior in response to directional shifts in delay distributions (i.e., value up- and downshifts) to maximize the rate of reward acquisition, while maintaining stable choice preferences despite variability in outcomes under baseline conditions.

We note that the changes in option preference in response to shifts on HV and LV options were asymmetric: greater variance of outcome distribution facilitated behavioral adaptations in response to value upshifts; conversely, low expected outcome uncertainty led to potentiated responses to downshifts. This effect may be explained by the hyperbolic nature of delay-discounting across species (*Freeman et al., 2009*; *Green et al., 2013*; *Hwang et al., 2009*; *Mazur and Biondi, 2009*; *Mitchell et al., 2015*; *Rachlin et al., 1991*). Specifically, the delays in our task were normally distributed, but the perceived value distributions may be skewed. Since the HV option produces a greater proportion of immediate or short-delayed rewards, and therefore more valuable outcomes, it may generally be easier for animals to detect upshifts on this option. These more immediate rewards may be more salient and/or more preferred. Conversely, during the downshifts as delays get longer, differences in waiting time become less meaningful and the LV option that produces more delays of similar value could promote faster learning about worsening of reward conditions.

Despite these effects of delays on outcome valuation, our results demonstrated that rats can learn about shifts in values even when outcomes are uncertain. We then directly assessed the uncertainty-induced neural adaptations within the BLA and OFC and investigated causal contributions of these brain regions to value learning and decision making under expected outcome uncertainty.

## The BLA and ventral OFC undergo distinct patterns of neuroadaptations in response to outcome uncertainty

One of the most difficult challenges faced by an animal learning in an unstable habitat is correctly distinguishing between true changes in the environment that require new learning from stochastic feedback under mostly stable conditions. Indeed, the problem of change-point detection has long been studied in relation to modulation of learning rates in RL and Bayesian learning theory (*Behrens et al., 2007*; *Courville et al., 2006*; *Dayan et al., 2000*; *Gallistel et al., 2001*; *Pearce and Hall, 1980*; *Pearson and Platt, 2013*; *Yu and Dayan, 2005*). Long-term neuroadaptations in response to experience with outcome uncertainty may benefit learning by altering signal-to-noise processing (*Hoshino, 2014*; *Liguz-Lecznar et al., 2015*; *Rössert et al., 2011*), such that only those surprising events that exceed the levels of expected variability in the environment produce neuronal responses and affect behavior.

We directly assessed the changes in expression of gephyrin (a reliable proxy for membrane-inserted GABA$_A$ receptors mediating fast inhibitory transmission; [*Chhatwal et al., 2005*; *Tyagarajan et al., 2011*]) and GluN1 (an obligatory subunit of glutamate NMDA receptors; [*Soares et al., 2013*]) in BLA and ventral OFC in three separate groups of animals following prolonged experience with low and high levels of expected uncertainty in outcome distribution. Both gephyrin and GluN1 showed robust uncertainty-dependent upregulation in the BLA that was maximal after experience with highly uncertain conditions. Conversely, within the ventral OFC, gephyrin was downregulated following reward experience in general and did not depend on the degree of uncertainty in outcomes. However, our experiments did not include a certain control group (i.e., animals receiving rewards following a predictable delay on all trials). Therefore, we cannot exclude the possibility that changes in protein expression in the OFC in response to reward experience required some, albeit small, levels of outcome uncertainty.

Adaptations to expected uncertainty at the protein level are likely to diminish responses to subsequent trial-by-trial surprise signals in BLA. Concurrent increases in the sensitivity to excitation and inhibition benefit signal-to-noise processing, providing further evidence in support of this view (*Hoshino, 2014*; *Liguz-Lecznar et al., 2015*; *Rössert et al., 2011*). To detect environmental changes, animals need to compare current prediction errors to the levels of expected outcome uncertainty. Previous work has shown that GABA-ergic interneurons in BLA gate the information

flow and determine the signal intensity that is passed to postsynaptic structures (*Wolff et al., 2014*). The intrinsic excitability of pyramidal neurons (*Motanis et al., 2014*; *Paton et al., 2006*) and activity of interneurons in the BLA are shaped by reward experiences, possibly via a dopamine-dependent mechanism (*Chu et al., 2012*; *Merlo et al., 2015*). Upregulation of functional GABA$_A$ receptors as suggested by our data may decrease sensitivity to surprising events when outcome variability is high even under mostly stable conditions, while increases in GluN1 could support learning facilitation when the environment changes. Several psychiatric conditions such as anxiety, schizophrenia, obsessive compulsive and autism spectrum disorders, share pathological uncertainty processing as a core deficit, manifesting as a preference for stable, certain outcomes (*Winstanley and Clark, 2016a*, *2016b*). Interestingly, recent studies have similarly implicated mutations in the *gephyrin* gene as risk for autism and schizophrenia (*Chen et al., 2014*; *Lionel et al., 2013*). Future research may address the role of this synaptic organizer in surprise-driven learning and decision making under uncertainty in animal models of these disorders.

Contrary to the pattern of neuroadaptations observed in BLA, gephyrin in the OFC was downregulated in response to reward mean, but not expected uncertainty. These changes in protein expression may leave OFC responsivity to noisy value signals intact or even amplified, suggesting that one of its normal functions is to encode the richness of the outcome distribution or expected uncertainty signal. Indeed, previous reports demonstrated that at least some subpopulations of OFC neurons carry expected uncertainty representations during option evaluation and outcome receipt (*Li et al., 2016*; *van Duuren et al., 2009*). Based on these findings we hypothesized that BLA and ventral OFC may play complementary, yet dissociable, roles in decision making and learning under uncertainty.

## Ventral OFC causally contributes to learning under expected outcome uncertainty

Lesions to the ventral OFC produced a pronounced behavioral impairment on our task. These animals failed to change their choice preference in response to the majority of shifts. Paradoxically, the results of computational modeling revealed that responsivity to surprising outcomes was facilitated in these rats. Specifically, performance of OFC-lesioned animals was best accounted for by the PH model, suggesting that while these animals still increased learning rates in response to surprise (i.e., absolute prediction errors), they were insensitive to expected outcome uncertainty. Due to the lack of prediction error scaling based on the variability in experienced outcomes, OFC-lesioned animals treated every surprising event as indicative of a fundamental change in the value distribution and updated their expectations, rendering trial-by-trial value representations noisier, preventing consistent changes in preference. Because the delay distributions encountered during baseline and shift conditions in our task partially overlapped, inability to ignore meaningless fluctuations in outcomes would lead to unstable choice behavior and attenuated learning.

Complementary analyses of win-stay/ lose-shift strategy provide further support for potentiated sensitivity to surprising feedback in these animals: increased responsivity to both wins and losses emerged following ventral OFC lesions. Note that increased reliance on this strategy is highly suboptimal under stochastic environmental reward conditions (*Faraut et al., 2016*; *Imhof et al., 2007*; *Worthy et al., 2013*). Furthermore, we observed significantly reduced $\beta$ (inverse temperature parameter in softmax decision rule) values in OFC-lesioned group, indicating a noisier choice process and decreased reliance on learned outcome values in these animals. These results are in agreement with previous findings demonstrating increased switching and inconsistent economic preferences following ventral OFC lesions in monkeys (*Walton et al., 2010*, *2011*). Similarly, lesions to ventromedial prefrontal cortex, encompassing the ventral OFC, in humans render subjects unable to make consistent preference judgments (*Fellows and Farah, 2003*, *2007*). Importantly, human subjects with OFC damage cannot distinguish between degrees of uncertainty (*Hsu et al., 2005*). Similarly, previous work has implicated this brain region in prediction of reward timing (*Bakhurin et al., 2017*). We directly addressed whether BLA and ventral OFC lesions alter animals' ability to form expectations about the expected uncertainty in timing of reward delivery on our task. Whereas the distributions of reward-seeking times in BLA-lesioned animals were indistinguishable from control animals' and the true delays, OFC-lesioned animals concentrated their reward port entries in the time interval corresponding to mean delays, suggesting that while these animals can infer the average outcomes, they fail to represent the variance (i.e., expected uncertainty). These

findings are consistent with emerging evidence that more ventromedial regions, unlike lateral, OFC may be critical for decision making involving outcome uncertainty, but not response inhibition or impulsive choice behavior as suggested previously (*Stopper et al., 2014*).

Although frequently framed as a deficit in inhibitory control (*Bari and Robbins, 2013*; *Dalley et al., 2004*; *Elliott and Deakin, 2005*), medial OFC lesions or inactivations induce analogous effects in probabilistic reversal learning tasks where surprising changes in the reward distribution occur against the background of stochastic outcomes during the baseline conditions. For example, a recent study in rodents systematically compared the contributions of five different regions of the frontal cortex to reversal learning (*Dalton et al., 2016*). The results revealed unique contributions of the OFC to successful performance under probabilistic, but not deterministic conditions. Intriguingly, inactivations of the medial OFC impaired both the acquisition and reversal phases, suggesting that this subregion might be critical for many types of reward learning under conditions of expected outcome uncertainty. Since our lesions also intruded on medial OFC, our present observations are in agreement with these findings and suggest that one of the normal functions of more ventromedial sectors of OFC might be to stabilize value representations by adjusting responses to surprising outcomes based on expected outcome uncertainty.

Similar to previous work demonstrating that the OFC is not required for acquisition of simple stimulus-outcome associations or for unblocking driven by differences in value when outcomes are certain and predictable (*Izquierdo et al., 2004*; *McDannald et al., 2011*, *2005*; *Stalnaker et al., 2015*), we observed intact performance in OFC-lesioned animals during training to respond for rewards. It has been previously proposed that the OFC may provide value expectations that can be used to calculate RPEs to drive learning under more complex task conditions (*Schoenbaum et al., 2011a*, *Schoenbaum et al., 2011b*). Although this initial proposal was based on findings after targeting more lateral OFC subregions, our observations are generally consistent with this view and add a nuanced perspective. Specifically, if the OFC is needed to provide expectations about the value to which the observed outcomes are then compared, lesions of this brain region may result in attenuated learning driven by violation of expectations. The results of computational modeling in our work revealed a reduction in learning rates in OFC-lesioned animals consistent with this account. Yet our data provide further evidence that the representations of expected outcomes in ventral OFC are not limited to a single-point estimate of value, but also include information about expected uncertainty of variability in outcomes. This would allow an animal not only to detect if the outcomes violate expectations, but also to assess whether such surprising events are meaningful and informative to the current state of the world. If such events are important, an animal will shift its behavior, but if they may have occurred by chance, choices should remain unchanged.

Finally, more recently it has also been suggested that the OFC represents an animal's current location within an abstract cognitive map of the task it is facing (*Chan et al., 2016*; *Schuck et al., 2016*; *Wilson et al., 2014*), particularly when task states are not signaled by external sensory information, but rather need to be inferred from experience. In our task, animals may similarly represent different conditions, stable environment vs. shifted value, as separate states. Attenuated learning may result from state misrepresentations, where an animal incorrectly infers that it is currently in a stable environment and persists with the previous choice policy, despite the shift in value. As has been reported recently, neuronal activity in the lateral OFC organizes the task space according to the sequence of behaviorally significant events, or trial epochs. Conversely, neural ensembles in more medial OFC do not track the sequence of the events, but instead segregate between states depending on the trial value (*Lopatina et al., 2017*). In our study, ventromedial OFC may be especially well-positioned to encode upshifts and downshifts in value on long timescales, and loss of this function could cause an inability to recover appropriate state representations at the time of option choice.

Taken together with previous findings, our results implicate the OFC in representing fine-grained value distributions, including the expected uncertainty in outcomes (that may be task state-dependent). Consequently, lacking access to the complex outcome distribution, animals with OFC lesions over-rely on the average cached value.

## Functionally intact BLA is required for facilitation of learning in response to surprise

Whereas OFC lesions produced a pronounced impairment in performance on our uncertainty task, whether alterations induced by BLA lesions lead to suboptimal behavior is less clear. These animals changed their option preference to a lesser degree on HV upshifts, but compensated by exaggerated adaptations to HV downshifts. More detailed analyses of session-by-session data revealed specific alteration in responses to surprising value shifts under HV, but not LV, conditions in this group. Consistent with attenuated learning observed in animals with BLA lesions, trial-by-trial performance in this group was best fit by a RW+expected uncertainty model, demonstrating a selective loss of learning rate scaling in response to surprise and preserved RPE scaling with expected outcome uncertainty, leading to slower value learning compared to intact animals during the HV upshift. Note that suboptimal performance during even two or three sessions in our task (each session lasting 60 trials) means that BLA-lesioned animals are less efficient at reward procurement for 120–180 experiences. In naturalistic settings, such an early-learning impairment can have detrimental consequences. In agreement with the results of computational modeling, BLA-lesioned animals were less likely to adopt the win-stay/lose-shift strategy compared to the control group, demonstrating decreased sensitivity to surprising outcomes.

Whereas the lack of learning facilitation can account for reduced changes in preference in response to HV upshifts in BLA-lesioned animals, it may seem at odds with potentiated responses to downshifts on this option. Our computational modeling results suggest that control animals potentiate their learning in response to highly surprising outcomes, which leads to greater behavioral adaptations in the first few sessions during the shifts. In BLA-lesioned animals, this function is lost, and learning proceeds at the same rate. This results in significantly reduced choice adaptations throughout the HV upshift sessions. Yet BLA-lesioned animals adapt much more to the downshift on HV option. This difference appears to be in the performance asymptote, as learning still progresses linearly in BLA-lesioned group. A couple of factors may drive this effect. Firstly, as discussed earlier hyperbolic discounting leads to a larger impact of short delays on behavior. Immediate or short-delayed rewards encountered during upshift on HV option will potentiate learning in control animals early on during the shift, but fail to do so in BLA lesioned animals. During the downshift on HV option, as delays get longer, differences in waiting times become less meaningful as there is a smaller effect of larger delays on perceived outcome values. Thus, learning will be potentiated, but only briefly in control animals, but will still proceed linearly in BLA-lesioned rats. Additionally, potentiated responses to downshifts on HV option in this group may result from uncertainty avoidance interacting with surprise-driven learning. Indeed, we observed a consistent increase in uncertainty aversion in BLA-lesioned animals. Our computational models did not include an explicit uncertainty avoidance parameter as we were primarily interested in exploring alterations in learning.

Previous findings have implicated the BLA in updating reward expectancies when the predictions and outcomes are incongruent and facilitating learning in response to surprising events (*Ramirez and Savage, 2007*; *Savage et al., 2007*; *Wassum and Izquierdo, 2015*). Indeed, predictive value learning in the amygdala involves a neuronal signature that accords with an RL algorithm (*Dolan, 2007*). Specifically, single-unit responses in the BLA correspond to the unsigned prediction error signals (*Roesch et al., 2010*) that are necessary for learning rate scaling in both RL and Bayesian updating models. The BLA utilizes positive and negative prediction errors to boost cue processing, potentially directing attention to relevant stimuli and potentiating learning (*Chang et al., 2012*; *Esber and Holland, 2014*) as demonstrated in downshift procedures with reductions in reward amount. These effects are frequently interpreted as surprise-induced enhancement of cue associability. Notably, a similar computational role for the amygdala has been proposed based on Pavlovian fear conditioning in humans, where cue-shock associations were also probabilistic, highlighting the general role for the amygdala in fine-tuning learning according to the degree of surprise (*Li et al., 2011*). Taken together, the accumulated literature suggests that this contribution of the BLA is apparent for both appetitive and aversive outcomes, for cues in different sensory modalities, and as we demonstrate here, the role is not limited to changes in outcome contingencies, but also supports learning about surprising changes in delay costs.

## BLA and OFC lesions induce uncertainty-avoidance

In addition to their effects on value learning, lesions to both the BLA and ventral OFC induced an uncertainty-avoidant phenotype with animals in both experimental groups demonstrating reduced preference for the HV option under baseline conditions compared to control group at the beginning of testing. Similarly, previous findings demonstrated that lesions or inactivations of the BLA shift the behavior away from uncertain options and promote choices of safer outcomes (*Ghods-Sharifi et al., 2009*; *Zeeb and Winstanley, 2011*). However, inactivations of the medial OFC have been shown to produce consistent shifts toward the uncertain option (*Winstanley and Floresco, 2016b*). Despite demonstrating pronounced risk-aversion at the beginning of the task, OFC-lesioned animals in our experiments progressively increased their preference for HV option with experience, suggesting that the effects on stable choice preference depend critically on the timing of OFC manipulations.

In summary, we show that both BLA and ventral OFC are causally involved in decision making and value learning under conditions of outcome uncertainty. Functionally-intact BLA is required for facilitation of learning in response to surprise, whereas ventral OFC is necessary for an accurate representation of outcome distributions to stabilize value expectations and maintain choice preferences.

## Materials and methods

Subjects were 56 naïve male outbred Long Evans rats (Charles River Laboratories, Crl:LE, Strain code: 006). All animals arrived at our facility at PND 70 (weight range 300–350 at arrival). Vivaria were maintained under a reversed 12/12 hr light/dark cycle at 22°C. Rats were left undisturbed for 3 days after arrival to our facility to acclimate to the vivarium. Each rat was then handled for a minimum of 10 min once per day for 5 days. Animals were food-restricted to ensure motivation to work for food for a week prior to and during the behavioral testing, while water was available *ad libitum*, except during behavioral testing. All animals were pair-housed at arrival and separated on the last day of handling to minimize aggression during food restriction. We ensured that animals did not fall below 85% of their free-feeding body weight. On the two last days of food restriction prior to behavioral training, rats were fed 20 sugar pellets in their home cage to accustom them to the food rewards. All behavioral procedures took place 5 days a week between 8am and 6pm during the rats' active period. Because we utilized a novel decision making task, we did not use an a priori power analysis to determine sample size for initial cohort of naïve animals. The chosen group size (n = 8) is consistent with previous reports in our lab. For subsequent behavioral experiments with sham, OFC, or BLA lesions we determined the animal numbers using an *a priori* sample size estimation for F test family in G*Power 3.1 (http://www.gpower.hhu.de/en.html). The analyses were based on the variance parameters obtained in the pilot experiments (reported in *Figure 1* and associated *Figure 1—source data 1*) and the number of independent variables as well as the interactions of interest in planned analyses. The analysis yielded a projected minimum of 7–8 animals per group when no surgical procedures are required. However, considering the possibility of surgical attrition, we set n = 8 per group. Research protocols were approved by the Chancellor's Animal Research Committee at the University of California, Los Angeles.

### Behavioral training

Behavioral training was conducted in operant conditioning chambers (Model 80604, Lafayette Instrument Co., Lafayette, IN) that were housed within sound- and light- attenuating cubicles. Each chamber was equipped with a house light, tone generator, video camera, and LCD touchscreen opposing the pellet dispenser. The pellet dispenser delivered 45 mg dustless precision sucrose pellets. Software (ABET II TOUCH; Lafayette Instrument Co., Model 89505) controlled the hardware. All testing schedules were programmed by our group and can be requested from the corresponding author. During habituation, rats were required to eat five sugar pellets out of the dispenser inside of the chambers within 15 min before exposure to any stimuli on the touchscreen. They were then trained to respond to visual stimuli presented in the central compartment of the screen within 40 s time interval in order to receive the sugar reward. During the next stage of training, animals learned to initiate the trial by nose-poking the bright white square stimulus presented in the central compartment of the touchscreen within 40 s; this response was followed by disappearance of the central

stimulus and presentation of a target image in one of the side compartments of the touchscreen (immediately to the left or right of the initiation stimulus). Rats were given 40 s to respond to the target image, which was followed by an immediate reward. The last stage of training was administered to familiarize animals with delayed outcomes. The protocol was identical to the previous stage, except the nosepoke to the target image and reward delivery were separated by a 5 s stable delay. Across all stages of pre-training, failure to respond to a visual stimulus within the allotted time resulted in the trial being scored as an omission and beginning of a 10 s ITI. All images used in pre-training were pulled from the library of over 100 visual stimuli and were never the same as the images used in behavioral testing described below. This was done to ensure that none of the visual cues acquired incentive value that could affect subsequent performance. Criterion for advancement into the next stage was set to 60 rewards collected within 45 min.

## Behavioral testing

Task design and behavior of intact animals are illustrated in *Figure 1*, *Video 1* and *Video 2*. Our task is designed to assess the effects of expected outcome uncertainty on learning. We have elected to focus on reward rate (outcome value was determined by delay to reward receipt) rather than reward magnitude to avoid the issue of satiety throughout the testing session. Each trial began with stimulus (bright white square) presentation in the central compartment of the touchscreen. Rats were given 40 s to initiate a trial. If 40 s passed without a response, the trial was scored as an 'initiation omission'. Following a nosepoke to the central compartment, the central cue disappeared and two choice stimuli were presented concurrently in each of the side compartments of the touchscreen allowing an animal a free choice between two reward options. In our task stimulus-response side assignments were held constant for each animal to facilitate learning. Side-stimulus assignments were counterbalanced across animals, and held constant between sessions. Each response option was associated with the delivery of one sugar pellet after a delay interval. The delays associated with each option were pooled from distributions that are identical in mean value, but different in variability (LV vs HV; $\sim N(\mu, \sigma)$: $\mu = 10$ s, $\sigma_{HV}=4$s $\sigma_{LV}=1$s). An animal was given 40 s to make a choice; failure to select an option within this time interval resulted in the trial being scored as 'choice omission' and beginning of an ITI.

Therefore, rats were presented with two options identical in mean (10 s) but different in the variance of the delay distribution (i.e., expected outcome uncertainty). Following the establishment of stable performance (defined as no statistical difference in any of the behavioral parameters across three consecutive testing sessions), rats experienced reward upshifts (delay mean was reduced to 5 s with variance kept constant) and downshifts (20 s) on each option independently, followed by return to baseline conditions. Thus, in upshifts rats were required to wait less on average for a single sugar pellet, whereas in downshifts, rats were required to wait longer, on average. The order of shift experiences was counterbalanced across animals. Animals were given one testing session per day that was terminated when an animal had collected 60 rewards or when 45 min had elapsed. Each shift and return to baseline phase lasted for five sessions. Therefore, rats experienced a total number of 43 sessions with varying delays. We first trained a group of naïve rats (n = 8) on this task to probe the ability to distinguish true changes in the environment from stochastic fluctuations in outcomes under baseline conditions in rodents. The animals in lesion experiments (n = 24: n sham = 8, n BLA lesion = 8; n OFC lesion = 8) were tested under identical conditions. Each animal participated in a single experiment. For each experiment, rats were randomly assigned into groups.

## Protein expression analyses

Three separate groups of animals were trained to respond to visual stimuli on a touchscreen to procure a reward after variable delays. The values of outcomes were identical to our task described above but no choice was given. One group was trained under LV conditions, the second under HV (matched in total number of rewards received), and the third control group received no rewards (n = 8 in each group; total n = 24). The training criterion was set to a 60 sugar pellets for three consecutive days to mimic the baseline testing duration in animal trained on our main task. Rats were euthanized 1d after the last day of reward experience with an overdose of sodium pentobarbital (250 mg/kg, i.p.) and decapitated. The brains were immediately extracted and two mm-thick coronal sections of ventral OFC and BLA were further rapidly dissected, using a brain matrix, over wet ice at

4°C. To prepare the tissues for the assays 0.2 mL of PBS (0.01 mol/L, pH 7.2) containing a protease and phosphatase inhibitor cocktail (aprotinin, bestatin, E-64; leupeptin, NaF, sodium orthovanadate, sodium pyrophosphate, $\beta$-glycerophosphate; EDTA-free; Thermo Scientific, Rockford, IL; Product # 78441) was added to each sample. Each tissue was minced, homogenized, sonicated with an ultrasonic cell disrupter, and centrifuged at 5000 g at 4°C for 10 min. Supernatants were removed and stored at +4°C until ELISA assays were performed (within 24 hr period). Bradford protein assays were also performed to determine total protein concentrations in each sample. The assays were performed according to the manufacturer's instructions. The sensitivity of the assays is 0.1 ng/ml for gephyrin (Cat# MBS9324933) and GluN1 (Cat# MBS724735, MyBioSource, Inc, San Diego, CA) and the detection range is 0.625 ng/ml - 20 ng/ml. The concentration of each protein was quantified as ng/mg of total protein accounting for dilution factor and presented as percent of no reward group.

## Surgery

Excitotoxic lesions of BLA (n = 8) and ventral OFC (n = 8) were performed using aseptic stereotaxic techniques under isoflurane gas (1–5% in O2) anesthesia prior to behavioral testing and training. Before surgeries, all animals were administered 5 mg/kg s.c. carprofen (NADA #141–199, Pfizer, Inc., Drug Labeler Code: 000069) and 1cc saline. After being placed into a stereotaxic apparatus (David Kopf; model 306041), the scalp was incised and retracted. The skull was then leveled to ensure that bregma and lambda are in the same horizontal plane. Small burr holes were drilled in the skull to allow cannulae with an injection needle to be lowered into the BLA (AP: −2.5; ML: ± 5.0; DV: −7.8 (0.1 µl) and −8.1 (0.2 µl) from skull surface) or OFC (0.2 µl, AP =+3.7; ML = ±2.0; DV = −4.6). The injection needle was attached to polyethylene tubing connected to a Hamilton syringe mounted on a syringe pump. N-Methyl-D-aspartic acid (NMDA, Sigma-Aldrich; 20 mg/ml in 0.1 m PBS, pH 7.4; Product # M3262) was bilaterally infused at a rate of 0.1 µl/min to destroy intrinsic neurons. After each injection, the needle was left in place for 3–5 min to allow for diffusion of the drug. Sham-lesioned group (n = 8) underwent identical surgical procedures, except no NMDA was infused. All animals were given one-week recovery period prior to food restriction and subsequent behavioral testing. During this week, the rats were administered 5 mg/kg s.c. carprofen (NADA #141–199, Pfizer, Inc., Drug Labeler Code: 000069) and their health condition was monitored daily.

## Histology

The extent of the lesions was assessed by staining for NeuN, a marker for neuronal nuclei. After the termination of training, animals were sacrificed by pentobarbital overdose (Euthasol, 0.8 mL, 390 mg/mL pentobarbital, 50 mg/mL phenytoin; Virbic, Fort Worth, TX) and transcardial perfusion. Brains were post-fixed in 10% buffered formalin acetate for 24 hr followed by 30% sucrose for 5 days. Forty µm coronal sections containing the OFC and BLA were first incubated for 24 hr at 4°C in solution containing primary NeuN antibody (Anti-NeuN (rabbit), 1:1000, EMD. Millipore, Cat. # ABN78), 10% normal goat serum (Abcam, Cambridge, MA, Cat. # ab7481), and 0.5% Triton-X (Sigma, St. Louis, MO, Cat. # T8787) in 1X PBS, followed by three 10 min washes in PBS. The tissue was then incubated for 4 hr in solution containing 1X PBS, Triton-X and a secondary antibody (Goat anti-Rabbit IgG (H+L), Alexa Fluor 488 conjugate, 1:400, Fisher Scientific, Catalog #A-11034), followed by three 10 min washes in PBS. Slides were subsequently mounted and cover-slipped, visualized using a BZ-X710 microscope (Keyence, Itasca, IL), and analyzed with BZ-X Viewer software. Lesions were determined by comparison with a standard rat brain atlas (*Paxinos and Watson, 1997*).

## Computational analyses

We fit different versions of reinforcement learning models to trial-by-trial choices for each animal separately. Specifically, we considered the standard Rescorla-Wagner model (RW) and a dynamic learning rate model (Pearce-Hall, PH). Trials from all sessions were treated as contiguous. Option values were updated in response to RPE, $\delta_t$, weighted by the learning rate, $\alpha$ (constrained to the interval [0 1]). The expected value for each option was updated according to delta rule:

$$Q_{t+1} \leftarrow Q_t + \alpha^\star \delta_t.$$

The $\delta_t$ is the difference between current outcome $V_t$ and expected value $Q_t$. Given that the value

of each outcome was determined by delay to reward of a constant magnitude, $V_t$ was specified as $1/(1-kD)$, where D is the duration of delay and $k$ $[0, +\infty]$ is a free parameter setting the steepness of the discounting curve. In dynamic learning rate models (PH and PH+expected uncertainty described below), $\alpha$ was updated in response to the degree of surprise (absolute $\delta_t$) according to:

$$\alpha_{t+1} \leftarrow |\delta_t|^\star \eta + (1-\eta)^\star \alpha_t.$$

We set initial $\alpha$ for HV and LV options to the same value, but allowed independent updating with experience. We also considered models in which expected outcome uncertainty is learned simultaneously with value and scales the impact of prediction errors on value (RW+expected uncertainty) and learning rate (Full model) updating. Uncertainty prediction errors are the difference between squared expected and realized RPEs. Expected uncertainty expectations are subsequently updated according to delta rule. Therefore, in the Full model:

$$Q_{t+1} \leftarrow Q_t + \alpha_t {}^\star \delta_t / \omega^\star \exp(\sqrt{\sigma'_t});$$

where $\omega$ $[1, +\infty]$ is a free parameter determining individual sensitivity to expected uncertainty.

$$\alpha_{t+1} \leftarrow \eta^\star |\delta_t| / \omega^\star \exp(\sqrt{\sigma'_t}) + (1-\eta)^\star \alpha_t.$$
$$\alpha_{t+1}' \leftarrow \sigma_t' + \alpha_{\mathrm{risk}}{}^\star \delta_{\mathrm{risk},t}; \delta_{\mathrm{risk,t}} = \delta_t^2 - \delta_t'$$

The option choice probability on each trial was determined according to a softmax rule with an inverse-temperature parameter $\beta$; $\propto \exp(\beta^\star Q_t)$.

The model parameters were estimated to maximize probability of obtaining the observed vector of choices given the model and its parameters (by minimizing negative log likelihood computed based on the difference between predicted choice probability and the actual choice on each trial using *fmincon* in MatLab). We used Bayesian information criterion (BIC) instead of AIC as a more conservative measure to determine the best model. The total number of free parameters, BIC and parameter values for each model and experimental group are provided in *Table 1*.

## Behavioral and statistical analyses

Software package SPSS (SAS Institute, Inc., Version 24) and MatLab (MathWorks, Natick, Massachusetts; Version R2016b) were used for statistical analyses. Statistical significance was noted when p-values were less than 0.05. Shapiro Wilk tests of normality, Levene's tests of equality of error variances, Box's tests of equality of covariance matrices, and Mauchly's tests of sphericity were used to characterize the data structure.

Protein expression data were analyzed with univariate ANOVA with reward experience group (HV, LV, or no reward) as the between-subject factor. Maximal changes in choice of each option in response to shifts were analyzed with omnibus ANOVA with shift type (HV, LV; upshift, downshift) and shift phase (pre-baseline, shift, post-baseline) as within-subject factors (total number of animals, n, in this analysis = 8). Similar analyses were performed on data obtained from lesion experiments with an additional between-subject factor of experimental group (sham, BLA vs OFC lesion; total n = 24, n = 8 per group). Furthermore, we subjected the session-by-session data during each swift to an omnibus ANOVA with testing session (1 through 5) and shift type as within- and experimental group as between subject factors.

### Win-stay/Lose-shift

To evaluate whether the animals in lesioned groups adopted a different strategy and demonstrated altered sensitivity to surprising outcomes, we examined the win-stay/lose-shift response strategy. Win-stay/lose-shift score was computed based on trial-by-trial data similar to previous reports (*Faraut et al., 2016*; *Imhof et al., 2007*; *Worthy et al., 2013*). The algorithm used for this analysis kept track of all delays experienced before the current trial under baseline conditions for each animal individually. On each trial, we calculated the mean of the experienced baseline delay distribution and found the value of the minimal and maximal delay. If the current delay value fell within this interval (i.e., min prior delay $\leq$ current delay $\geq$ max prior delay), the outcome was classified as *expected*. If the current delay fell outside of this distribution (current delay $\leq$ min prior delay or current delay $\geq$ max prior delay), the outcome on this trial was classified as *unexpected* (surprising). Trials on which

the current delay exceeded the mean of experienced delay distribution were counted as *wins* and delays lower than the mean were classified as *losses*. We counted rats' decisions as *stays* when they chose the same option on the subsequent trial and as *shifts* when the animals switched to the other alternative. Therefore, each trial could be classified as *win-stay*, *win-shift*, *lose-stay*, or *lose-shift*. Win-stay and lose-shift trials were given a score of 1 and win-shifts and lose-stays were counted as 0 s. We considered all baseline and value-shift trials; however, trials with delays equal to the mean of previously experienced distribution or trials followed by choice omissions were excluded from this analysis. Win-stay and lose-shift scores we calculated for each trial type separately and their probabilities (summary score divided by the number of trials) for both trial types were subjected to ANOVA with strategy as within-subject and experimental group as between-subject factors.

### Reward port entries

To gain further insights into outcome representations in our experimental groups, we addressed whether BLA and ventral OFC lesions altered animals' ability to form expectations about the timing of reward delivery. On each trial during all baseline conditions, where the overall values of LV and HV options were equivalent, reward port entries were recorded during the waiting period. This analysis included all trials under initial baseline conditions and baselines separating the shifts. Since reward delivery in our task was signaled to animals by illumination of the magazine and sounds made by the dispenser and pellet drop, rats generally collected rewards immediately (median reaction time from reward delivery to consumption = 0.84 s). Because our aim was to assess outcome expectations, rather than reactions to reward delivery, we only analyzed the time interval starting at disappearance of visual stimuli following the choice and terminating at the end of the delay period (magazine entries after the pellet has been dispensed were excluded from this analysis). The waiting period was split into 1 s bins and all magazine entries were recorded in each interval. We then divided the number of entries in each bin by the total number of entries to obtain probabilities. These data we analyzed with multivariate ANOVA with option (LV, HV) and time bin as within- and experimental group as between-subject factors. Mauchly's tests of sphericity were used to compare variances across groups.

When significant interactions were found, post hoc simple main effects were reported. Dunnett t (2-sided) comparisons were applied when assessing the differences between experimental and a single control groups, whereas Bonferroni correction was applied to multiple comparisons. Where the assumptions of sphericity were violated, Greenhouse-Geisser p-value corrections were applied (Epsilon <0.75). Group mean values and associated SEM are reported in figures (individual data are provided in Source_Data files).

## Acknowledgements

This work was supported by UCLA's Division of Life Sciences Recruitment and Retention fund and UCLA Opportunity Fund (Izquierdo). We are grateful to Jose Gonzalez for his assistance with data collection, as well as Evan Hart and members of Lau lab for their valuable input.

## Additional information

### Funding

| Funder | Author |
| --- | --- |
| UCLA Division of Life Sciences Recruitment and Retention Fund | Alicia Izquierdo |
| UCLA Opportunity Fund | Alicia Izquierdo |

The funders had no role in study design, data collection and interpretation, or the decision to submit the work for publication.

## Author contributions

AS, Conceptualization, Data curation, Formal analysis, Investigation, Visualization, Methodology, Writing—original draft, Writing—review and editing; AI, Conceptualization, Data curation, Formal analysis, Supervision, Funding acquisition, Visualization, Methodology, Writing—original draft, Writing—review and editing

## Author ORCIDs

Alexandra Stolyarova, http://orcid.org/0000-0003-4397-4895

Alicia Izquierdo, http://orcid.org/0000-0001-9897-2091

## Ethics

Animal experimentation: This study was performed in strict accordance with the recommendations in the Guide for the Care and Use of Laboratory Animals of the National Institutes of Health. Research protocols (#2013-094-13A) were approved by the Chancellor's Animal Research Committee at the University of California, Los Angeles. All surgeries were performed under isoflurane anesthesia, and every effort was made to minimize suffering.

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
