## [Decision Letter]

Thank you for submitting your article "Complementary contributions of basolateral amygdala and orbitofrontal cortex to value learning under uncertainty" for consideration by *eLife*. Your article has been favorably evaluated by Timothy Behrens (Senior Editor) and two reviewers, one of whom, Geoffrey Schoenbaum (Reviewer #1), is a member of our Board of Reviewing Editors.

The reviewers have discussed the reviews with one another and the Reviewing Editor has drafted this decision to help you prepare a revised submission.

Quick note from the Senior Editor.

– Just a quick note to say that myself and two other editors looked at this at triage stage and we all independently raised the point the Reviewing Editor is making at the end of the essential revisions section about the framing of the paper. It seemed to us that the task does not get at expected vs. unexpected uncertainty, or at volatility, but rather at learning in the context of different known variances. See the Reviewing Editor’s point below.

Second, the early results are a bit confusing, as high and low variance conditioning have opposing effects on an upward and downward shift. It is possible that this is partly due to the fact that you induce the variance by changing the delay, but do so in a linear way. The effect of delay on value is presumably hyperbolic in rats as in other species, so the value distribution is presumably skewed (not Gaussian) and differentially skewed depending on the variance, causing differential choice sensitivity on up-vs. down-shift in these two situations. I am not asking for more data, but it seems like something that might be acknowledged.

From the Reviewing editor.

Summary:

In this study the authors develop a novel task in which they can examine the impact of different levels of uncertainty in the distribution of the timing of reward on learning about subsequent changes in that distribution. The task is very nice. And they report novel effects of manipulations of OFC and BLA on learning in response to changes in high versus low variance distributions. Both reviewers thought the study was generally well designed, addressed an important question, and generated results that were novel and interesting. There were only a few significant issues, which seem addressable.

Essential revisions:

In our discussions, there seemed to be several points in which we were in agreement. First both reviewers felt that the immunohistochem in Figure 2 and the dopamine manipulations at the end could be eliminated or moved to the supplemental. In the case of Figure 2, this was mostly because the results were relatively weakly constrained and did not add much to the conclusions and so are not necessary. In the case of the dopamine data, it was more that they did not fit with the study. In each case, the impact of the main findings will be strengthened by pruning these if possible.

A second major issue was that we would like to see the actual performance data, especially in Figure 6, rather than representation of changes. We appreciate if the data is easier to grasp as a change, so it is fine to keep this. But how the underlying raw behavior changes to produce these shifts may be important so this should be represented.

A third major issue was the lack of any illustration of the lesions. Please show a photomic of the lesion and clarify coordinates and extent of lesions.

Finally there was some concern over the framing (mostly by me). I think the paper is extremely elegant and interesting. But I do not see the comparison as between certain and uncertain uncertainty, since the HV and LV distributions are both well-learned (and thus certain uncertainty) and the influence of uncertain uncertainty is not on performance but rather on learning. So really the proper framing it seems to me is in the effect of different levels of certain uncertainty on learning. As I note below, this is an extremely interesting question given the respective proposals about OFC and BLA in representing outcomes and driving associability. And given the relative lack of effects of manipulations of both areas on learning (unblocking/blocking) when the attributes (timing and the rest) of outcomes are certain or deterministic. I think the authors can avoid confusing reviewers and make clearer points if they consider reframing things this way. Or at a minimum, they must help me understand what is certain and uncertain about their uncertainty?

*Reviewer #1:*

In this study, the authors devise a novel task to distinguish effects of what they term expected uncertainty and unexpected uncertainty on behavior in a simple free-choice touchscreen task. Rats chose between two symbols on each trial to obtain a sugar reward. Rewards were delivered after an average delay, but one symbol indicated a high variance delay and the other a low variance delay (expected uncertainty). Once trained on this, rats were presented with sessions in which the distribution of delays for one or the other cue was shifted earlier or later (unexpected uncertainty). The authors show that the shifts cause changes in choice behavior, and interestingly the shifts are asymmetric, with upshifts causing a larger change in behavior when applied to the high variance distribution and downshifts causing a larger change in behavior when applied to the low variance distribution. They then probe effects of medial OFC or BLA lesions on performance in the task. They find effects but they differ with OFC lesioned rats showing little or no apparent effect of the surprising shifts in distributions, whereas BLA lesioned rats showing a more mixed pattern of impairments and enhancements. Combining analyses of changes in choice preference, win-stay/lose-shift behavior, modeling, and the timing of reward port behavior, the authors argue that the BLA deficit is due to a loss of attentional effects on learning rate from surprise (PH mechanism) whereas the OFC deficit was more related to a failure to learn and exploit appropriately if I followed the explanation correctly.

Overall I thought this was an extremely interesting study. The task is exceptionally novel and creative in my opinion, and it addresses a very interesting question of how uncertainty in the timing (and by extrapolation other features) of rewards encountered in the real world must interact with more global error signaling mechanisms that are linked to learning/updating our ideas about the world, either directly or via constructs such as attention or surprise. The results are very cool, and I think the authors’ interpretation of them is quite clever. I might ask for some clarifications, and I have a few suggestions to consider, but overall I think this is an excellent contribution to the literature.

I have a couple suggestions for the authors to consider. One concerns their framing. They talk about different forms of uncertainty, which I agree they do induce (though it is unclear what the boundary conditions are between them). However their task confounds this distinction with effects of uncertainty on learning. And ultimately their data is more about how the distributions impacted the learning response as opposed to how the rats use one type versus another for making decisions. This disconnection caused me some angst initially because I was expecting them to do something to violate the distributions and look for some immediate effect on behavior – execute an abort option, move to another patch, or something. These are things that people do study of course. What the authors are doing here is very different I think. The authors may wish to consider whether there is a small or large change that can be made to the Introduction to make clearer what is being studied.

In this regard, I think the only major concern I can find with the study is that it does not include a no variance option. There are studies that show these areas are often not necessary for simple conditioning and discrimination tasks. This includes changes in learning in blocking and unblocking tasks in response to surprise. However these studies all use tasks where reward is 100% certain I think. I won't ask the authors to add this condition, as that would require repeating much of a very nice study. But I wonder if it is worth mentioning any of this work. In some cases, there will be location-of-lesion confounds. And the shifts are typically not done by changing timing. But even with these alternative interpretations, it might be worth noting where prior studies have found entirely negative effects with certain reward relationships. I think there are some by Holland, though I guess these are typically central nucleus. However I think more recently he has some work looking at basolateral amygdala maybe. And with regard to the orbitofrontal cortex, value blocking and unblocking is not affected by lesions or inactivation typically (Schoenbaum/McDannald).

I also think some addition to the Discussion to spell out precisely how the claimed functional losses would lead to the deficits observed would also be helpful – helping readers translate the modeling conclusions into explanations for the effects in Figure 4. For example, the downshift gets larger after BLA lesions. Helping readers puzzle out the relationships will make the conclusions stronger.

I also am unclear what is lost after OFC lesions. It seemed like a variety of things were described in the subsection “Complementary contributions of the BLA and OFC to value learning under uncertainty revealed by computational modeling”. But I think the win-stay/lose-shift performance and the data in Figure 5 if I understand them suggests a root problem maybe a loss of a fine-grained representation of the distributions. Given that this distribution is essentially an internal model of unobservable external states, this seems consistent with recent proposals that OFC maps task space, especially when not directly cued or observable (Niv). The historical distribution of prior reward times seems to fall into this category. It is also worth noting that this idea was developed to explain effects of OFC lesions on dopamine error signals, so it would predict some of these effects I think.

*Reviewer #2:*

This manuscript centres on the effects of ventromedial orbitofrontal and basolateral amygdala lesions on a novel timing choice task. Rats were trained to select between stimuli that initially were associated with equal mean but different variance in time of reward delivery (low v. high variance). They then sequentially experienced either increases or decreases in the mean reward time of the HV or LV option. A first study showed that shifts in value were associated with increases in GluN1 and a proxy for GABA A receptors in BLA and a reduction in GABA A in OFC (though against a rather weak "no reward" control). Of more interest, both behavioural and computational analyses showed dissociable effects of lesions to BLA or ventromedial OFC on choice performance following a shift. OFC lesioned animals had attenuated response to LV shifts and also were shown to respond selectively to the reward port at the mean reward delivery time. BLA animals, by contrast, retained the reward distribution and showed little impairment following LV shifts, but showed distinct changes following shifts in the HV. Modelling of the behaviour supported a selective loss of information about the reward time distribution in OFC-lesioned animals and a loss of learning rate scaling after BLA lesions.

I like a lot of things about this study. The task is novel and neat, the lesion effects, at least as presented, seem clear and distinct, and the modelling adds rigour to the authors' explanation. I had a couple of quibbles/questions about the analysis and explanation and would have liked a few more details to be presented, as detailed below. I also felt that the dopamine pharmacology at the end, while potentially interesting, didn't really add and felt out of place – I would drop this. Main additional points:

1) I understand why the authors chose to present the lesion data as a change from baseline, particularly given the data presented in Figure 6 showing baseline changes (at least initially) in the lesion groups. However, I also worried that it might potentially mask potentially important information. Could the lesion data be presented like Figure 1/D, at least in a supplementary figure, to enable more direct comparison to the actual performance?

I thought the analysis in Figure 4 was potentially interesting, but I wasn't sure I fully understood how it had been produced. The legend describes that a win-stay for a better than average or lose-switch for worse than expected was scored as a "1". Therefore, does a score of ~0.15 for sham win-stay expected mean that they were strongly likely to alternate (i.e., a win-switch on the majority of trials)? Some more details in the Materials and methods would be good.

2) Were the animals trained to baseline levels pre-surgery or did surgery happen first and then all training second? If the latter, it becomes particularly important to present the training to baseline data. While both are interesting, it is quite different if the effect is about learning the distribution of reward times compared to representing them, which the Abstract states.

3) I couldn't follow the shading scheme for the lesion figure in Figure 3 as there seemed to be only dark and less dark, not multiple gradations. A key at least should be added. Also, the OFC lesions also looked consistently medial to the stated injection site (2mm ML). Are these the correct coordinates? Can the authors also present photomicrographs of a sample lesion?

4) I thought the reward time distribution analyses were potentially interesting. Three questions: (1) it is described that there is a change in baseline performance in the OFC group over sessions. Is this at all reflected in the response time distribution or is this independent? (2) is the total number of head entries (i.e., a proxy of expected value) similar in the sham and lesion groups? (3) does the distribution also stay as concentrated around the mean in the OFC group following shifts in value?

---

## [Author Response]

Quick note from the Senior Editor.

*Just a quick note to say that myself and two other editors looked at this at triage stage and we all independently raised the point the Reviewing Editor is making at the end of the essential revisions section about the framing of the paper. It seemed to us that the task does not get at expected vs. unexpected uncertainty, or at volatility, but rather at learning in the context of different known variances. See the Reviewing Editor’s point below.*

We thank the Editor and reviewers for this helpful suggestion. We agree that our task is more straightforward to interpret as assessing learning under conditions of different levels of expected outcome uncertainty (i.e., outcome variances). We have introduced substantial changes to the Introduction and Discussion to re-frame our manuscript.

*Second, the early results are a bit confusing, as high and low variance conditioning have opposing effects on an upward and downward shift. It is possible that this is partly due to the fact that you induce the variance by changing the delay, but do so in a linear way. The effect of delay on value is presumably hyperbolic in rats as in other species, so the value distribution is presumably skewed (not Gaussian) and differentially skewed depending on the variance, causing differential choice sensitivity on up-vs. down-shift in these two situations. I am not asking for more data, but it seems like something that might be acknowledged.*

We agree that the asymmetry in behavioral adaptations to value shifts on HV and LV option could be explained by hyperbolic delay discounting. We had only briefly mentioned this point in the previous version of our manuscript, but in the revised version we added an extensive discussion (Discussion section).

*Essential revisions:*

*In our discussions, there seemed to be several points in which we were in agreement. First both reviewers felt that the immunohistochem in Figure 2 and the dopamine manipulations at the end could be eliminated or moved to the supplemental. In the case of Figure 2, this was mostly because the results were relatively weakly constrained and did not add much to the conclusions and so are not necessary. In the case of the dopamine data, it was more that they did not fit with the study. In each case, the impact of the main findings will be strengthened by pruning these if possible.*

We thank the reviewers for these suggestions. We agree that the dopamine data do not fit well with the main question investigated in our experiments, therefore, these data have been excluded from the present version.

However, we do believe that the gephyrin and GluN1 data are novel and informative. The changes in protein expression induced by experience with different levels of uncertainty are not examined frequently, and changes in gephyrin and GluN1, specifically, are underexplored. Yet, there is increasing evidence that “several psychiatric conditions such as anxiety, schizophrenia, obsessive compulsive and autism spectrum disorders, share pathological uncertainty processing as a core deficit, manifesting as a preference for stable, certain outcomes (Winstanley and Clark, 2016; Winstanley and Floresco, 2016). Interestingly, recent studies have similarly implicated mutations in the gephyrin gene as risk for autism and schizophrenia (Chen et al., 2014; Lionel et al., 2013)”. We think that the findings that we present here would appeal to broad *eLife* readership and could inform future pre-clinical work in disease models. For example, future research may address the role of gephyrin in surprise-driven learning and decision making under uncertainty in animal models of psychiatric disorders. We have discussed this in greater detail in our manuscript and also relate the findings presented in Figure 2 to the proposed roles of the BLA and OFC in learning under uncertainty in the third paragraph of the subsection “The BLA and ventral OFC undergo distinct patterns of neuroadaptations in response to outcome uncertainty”.

*A second major issue was that we would like to see the actual performance data, especially in Figure 6, rather than representation of changes. We appreciate if the data is easier to grasp as a change, so it is fine to keep this. But how the underlying raw behavior changes to produce these shifts may be important so this should be represented.*

These data have been added as a new Figure 4—figure supplement 1 and [Supplementary-material SD4-data]. We have also provided statistical analyses of these data: “In addition to examining the maximal changes in option preferences, we analyzed the behavioral data with an omnibus ANOVA with shift type and shift phase (pre-shift baseline, shift performance, and post-shift baseline) as within-subject and experimental group as between-subject factors. […] This pattern of results may be explained by changes in choice behavior even under baseline conditions in BLA- and OFC-lesioned animals that interacted with rats’ ability to learn about shifts in value”.

Also, based on this important suggestion, we changed the data presentation in Figure 6 to illustrate choice behavior in all baselines, not only the first and the last.

*A third major issue was the lack of any illustration of the lesions. Please show a photomic of the lesion and clarify coordinates and extent of lesions.*

We regret not including this in the first place. We have now changed Figure 3 and included a representative lesion for each experimental group. We also updated the Materials and methods section to include a more detailed NeuN staining protocol.

*Finally there was some concern over the framing (mostly by me). I think the paper is extremely elegant and interesting. But I do not see the comparison as between certain and uncertain uncertainty, since the HV and LV distributions are both well-learned (and thus certain uncertainty) and the influence of uncertain uncertainty is not on performance but rather on learning. So really the proper framing it seems to me is in the effect of different levels of certain uncertainty on learning. As I note below, this is an extremely interesting question given the respective proposals about OFC and BLA in representing outcomes and driving associability. And given the relative lack of effects of manipulations of both areas on learning (unblocking/blocking) when the attributes (timing and the rest) of outcomes are certain or deterministic. I think the authors can avoid confusing reviewers and make clearer points if they consider reframing things this way. Or at a minimum, they must help me understand what is certain and uncertain about their uncertainty?*

We thank this reviewer for his insightful comments. We agree that our task is more straightforward to interpret as assessing the effects of different levels of expected outcome uncertainty (i.e., outcome variances or certain uncertainty associated with each choice option) on learning, rather than animals’ ability to distinguish expected and unexpected uncertainty. We have incorporated substantial changes to the Introduction and Discussion to re-frame our manuscript. In the Introduction, we also mention the lack of impairments produced by BLA and OFC lesions when outcomes are certain and fully predictable.

*Reviewer #1: […] I have a couple suggestions for the authors to consider. One concerns their framing. They talk about different forms of uncertainty, which I agree they do induce (though it is unclear what the boundary conditions are between them). However their task confounds this distinction with effects of uncertainty on learning. And ultimately their data is more about how the distributions impacted the learning response as opposed to how the rats use one type versus another for making decisions. This disconnection caused me some angst initially because I was expecting them to do something to violate the distributions and look for some immediate effect on behavior – execute an abort option, move to another patch, or something. These are things that people do study of course. What the authors are doing here is very different I think. The authors may wish to consider whether there is a small or large change that can be made to the Introduction to make clearer what is being studied.*

We agree with this assessment. We have introduced substantial changes to the manuscript to re-frame the paper.

*In this regard, I think the only major concern I can find with the study is that it does not include a no variance option. There are studies that show these areas are often not necessary for simple conditioning and discrimination tasks. This includes changes in learning in blocking and unblocking tasks in response to surprise. However these studies all use tasks where reward is 100% certain I think. I won't ask the authors to add this condition, as that would require repeating much of a very nice study. But I wonder if it is worth mentioning any of this work. In some cases, there will be location-of-lesion confounds. And the shifts are typically not done by changing timing. But even with these alternative interpretations, it might be worth noting where prior studies have found entirely negative effects with certain reward relationships. I think there are some by Holland, though I guess these are typically central nucleus. However I think more recently he has some work looking at basolateral amygdala maybe. And with regard to the orbitofrontal cortex, value blocking and unblocking is not affected by lesions or inactivation typically (Schoenbaum/McDannald).*

We thank Dr. Schoenbaum for these literature suggestions. We have cited the relevant work in the revised manuscript.

*I also think some addition to the Discussion to spell out precisely how the claimed functional losses would lead to the deficits observed would also be helpful – helping readers translate the modeling conclusions into explanations for the effects in Figure 4. For example, the downshift gets larger after BLA lesions. Helping readers puzzle out the relationships will make the conclusions stronger.*

*I also am unclear what is lost after OFC lesions. It seemed like a variety of things were described in the subsection “Complementary contributions of the BLA and OFC to value learning under uncertainty revealed by computational modeling”. But I think the win-stay/lose-shift performance and the data in Figure 5 if I understand them suggests a root problem maybe a loss of a fine-grained representation of the distributions. Given that this distribution is essentially an internal model of unobservable external states, this seems consistent with recent proposals that OFC maps task space, especially when not directly cued or observable (Niv). The historical distribution of prior reward times seems to fall into this category. It is also worth noting that this idea was developed to explain effects of OFC lesions on dopamine error signals, so it would predict some of these effects I think.*

We have expanded the Discussion section to link the different types of analyses that we use in our experiments. We agree that the potentiated response to downshifts on HV in BLA-lesioned group might be puzzling and we provide several potential explanations for this effect. We note that the main prediction of the lack of PH-like learning rate scaling is linear learning. We have consistently observed potentiated learning during the first few shift sessions, followed by a plateau, in control animals. Yet in BLA-lesioned group learning curves appear to be always linear. Therefore, potentiated responses to HV downshift are likely due to changes in learning asymptote rather than slope. We believe that this change in performance asymptote may result from uncertainty aversion interacting with surprise-driven learning. Text has been added to Discussion to elaborate on this point in the second paragraph of the subsection “Functionally intact BLA is required for facilitation of learning in response to surprise”.

We also discussed relevant literature demonstrating similar contributions of the BLA across species in the third paragraph of the aforementioned subsection.

For OFC, we agree that the main impairment introduced by the lesions to this brain region appears to be the loss of fine-grained representations of outcome distributions. Some of the impairments could have also resulted from task state misrepresentations. We have now addressed these possibilities in the Discussion in the fourth and fifth paragraphs of the subsection “Ventral OFC causally contributes to learning under expected outcome uncertainty”.

*Reviewer #2:*

*[…] I like a lot of things about this study. The task is novel and neat, the lesion effects, at least as presented, seem clear and distinct, and the modelling adds rigour to the authors' explanation. I had a couple of quibbles / questions about the analysis and explanation and would have liked a few more details to be presented, as detailed below. I also felt that the dopamine pharmacology at the end, while potentially interesting, didn't really add and felt out of place – I would drop this.*

We thank the reviewer for these insightful comments and suggestions, we have excluded the dopamine data; we address the remaining points below.

*Main additional points:*

*1) I understand why the authors chose to present the lesion data as a change from baseline, particularly given the data presented in Figure 6 showing baseline changes (at least initially) in the lesion groups. However, I also worried that it might potentially mask potentially important information. Could the lesion data be presented like Figure 1/D, at least in a supplementary figure, to enable more direct comparison to the actual performance?*

We initially chose to present these data as changes in performance since we were primarily interested in learning. However, we agree that presenting the baseline data is equally informative, so we have now included additional plots to illustrate Figure 4 data as we have previously done in Figure 1. These data have been added as a new Figure 4—figure supplement 1 and [Supplementary-material SD4-data]. We have also provided statistical analyses of these data: “In addition to examining the maximal changes in option preferences, we analyzed the behavioral data with an omnibus ANOVA with shift type and shift phase (pre-shift baseline, shift performance, and post-shift baseline) as within-subject and experimental group as between-subject factors. […] This pattern of results may be explained by changes in choice behavior even under baseline conditions in BLA- and OFC-lesioned animals that interacted with rats’ ability to learn about shifts in value”.

*I thought the analysis in Figure 4 was potentially interesting, but I wasn't sure I fully understood how it had been produced. The legend describes that a win-stay for a better than average or lose-switch for worse than expected was scored as a "1". Therefore, does a score of ~0.15 for sham win-stay expected mean that they were strongly likely to alternate (i.e., a win-switch on the majority of trials)? Some more details in the Materials and methods would be good.*

We regret omitting a sufficiently detailed description of our analyses in the previous version. In the current version, we now include additional details in Materials and methods and Results:

“Win-stay/lose-shift score was computed based on trial-by-trial data similar to previous reports (Faraut et al., 2016; Imhof et al., 2007; Worthy et al., 2013). […] Win-stay and lose-shift scores we calculated for each trial type separately and their probabilities (summary score divided by the number of trials) for both trial types were subjected to ANOVA with strategy as within-subject and experimental group as between-subject factors”.

We have similarly briefly addresses potential reasons for low win-stay scores:

“Interestingly, sham and BLA-lesioned animals demonstrated low win-stay and lose-shift scores when trial outcomes were expected; these animals were more likely to shift after better than average outcomes and persist with their choices after worse outcomes. […] Additionally, when outcomes are relative stable and predictable, rats may be more sensitive to long-term reward history and rely less on the outcome of any one given trial”.

*2) Were the animals trained to baseline levels pre-surgery or did surgery happen first and then all training second? If the latter, it becomes particularly important to present the training to baseline data. While both are interesting, it is quite different if the effect is about learning the distribution of reward times compared to representing them, which the Abstract states.*

As the reviewer noted, all surgeries in our experiments took place prior to any exposure to the behavioral apparatus or training. We now include pre-training data in Figure 4—figure supplement 2 and provide statistical analyses of these data in Results:

“Despite these differences in responses to shifts in value, we did not observe any deficits in basic reward learning in either the BLA- or OFC-lesioned animals. […] Similarly, there were no group differences in their responses to the introduction of 5s delay interval during pre-training [F(2,21)=0.679, p=0.518] or the number of sessions to reach stable performance during the initial baseline phase of our uncertainty task [F92,21)=0.262, p=0.772; Figure 4—figure supplement 2]”.

*3) I couldn't follow the shading scheme for the lesion figure in Figure 3 as there seemed to be only dark and less dark, not multiple gradations. A key at least should be added. Also, the OFC lesions also looked consistently medial to the stated injection site (2mm ML). Are these the correct coordinates? Can the authors also present photomicrographs of a sample lesion?*

We have changed Figure 3 and also included representative lesions. We have also provided a more detailed staining protocol in Materials and methods. In terms of OFC lesions spreading more medially, we have been consistently observing this pattern of spread both with NMDA-induced lesion as well as AAV-mediated DREADD expression using the same OFC coordinates (+2ML). We also observe posterior spread in cases of BLA. We think that this may be due to regional cytoarchitecture; there could also be a difference in appropriate coordinates between rat strains. We also note that such spread is not always visible with cresyl violet or Nissl staining, which generally mask some lesion areas due to the presence of other cell types. Here, we assessed lesions with NeuN staining, which only marks intact neuronal nuclei, and is a more sensitive measure of lesion size.

*4) I thought the reward time distribution analyses were potentially interesting. Three questions: (1) it is described that there is a change in baseline performance in the OFC group over sessions. Is this at all reflected in the response time distribution or is this independent? (2) is the total number of head entries (i.e., a proxy of expected value) similar in the sham and lesion groups? (3) does the distribution also stay as concentrated around the mean in the OFC group following shifts in value?*

We thank the reviewer for these questions, we have included new analyses that we believe have strengthened the conclusions drawn from these data. For total number of head entries: “all groups were matched in the total number of reward port entries [F(2,21)=0.462, p=0.636]”. These results are illustrated in new Figure 5—figure supplement 1.

Based on the reviewer’s suggestion we have also considered the changes in waiting times across our task. “We calculated the variance of reward port entry times during each baseline (initial phase of the task and four baseline separating the shifts) for each animal. […] Importantly, since our analyses only included the waiting time prior to reward delivery, these results suggest that OFC-lesioned animals retain the ability to form simple outcome expectations based on long-term experience, yet their ability to represent the more complex outcome distributions is compromised”.

These analyses allowed us to address the changes in reward port times with task progression and following experience with shifts. However, we acknowledge that there is a possibility that OFC-lesioned animals display increased behavioral variability briefly after the shift. Unfortunately, these analyses require a large number of trials in our task, so we could not analyze the data from only few trials after shifts.